# POSTCAST: GENERALIZABLE POSTPROCESSING FOR PRECIPITATION NOWCASTING VIA UNSUPERVISED BLURRINESS MODELING

**Junchao Gong**[1,2]\*, **Siwei Tu**[2,3]\*, **Weidong Yang**[2,3]\*, **Ben Fei**[2,4]†, **Kun Chen**[2,3], **Wenlong Zhang**[2]
**Xiaokang Yang**[1], **Wanli Ouyang**[2], **Lei Bai**[2]†
[1]Shanghai Jiao Tong University [2]Shanghai Artificial Intelligence Laboratory
[3]Fudan University [4]The Chinese University of Hong Kong

## ABSTRACT

Precipitation nowcasting plays a pivotal role in socioeconomic sectors, especially in severe convective weather warnings. Although notable progress has been achieved by approaches mining the spatiotemporal correlations with deep learning, these methods still suffer severe blurriness as the lead time increases, which hampers accurate predictions for extreme precipitation. To alleviate blurriness, researchers explore generative methods conditioned on blurry predictions. However, the pairs of blurry predictions and corresponding ground truth need to be generated in advance, making the training pipeline of generative parts cumbersome and limiting the generality of generative models within blur modes that appear in training data. By rethinking the blurriness in precipitation nowcasting as a blur kernel acting on predictions, we propose an alternative postprocessing method to eliminate the blurriness without the requirement of training with the pairs of blurry predictions and corresponding ground truth. Specifically, we utilize blurry predictions to guide the generation process of a pre-trained unconditional denoising diffusion probabilistic model (DDPM) to obtain high-fidelity predictions with eliminated blurriness. A zero-shot blur kernel estimation mechanism and an auto-scale denoise guidance strategy are introduced to adapt the unconditional DDPM to any blurriness modes varying from datasets and lead times in precipitation nowcasting. Extensive experiments are conducted on 7 precipitation radar datasets, demonstrating the generality and superiority of our method. Our code is available at https://github.com/jasong-ovo/PostCast.

## 1 INTRODUCTION

Precipitation nowcasting, which mostly depends on radar echo data, plays a vital role in predicting local weather conditions for up to six hours (CLIMA & TE). Accurately predicting precipitation events is one of the core tasks in weather prediction. It could mitigate the socioeconomic impacts of extreme precipitation events and serve as a critical tool for transportation management, agricultural productivity, and other aspects. Hence, many excellent methods have been proposed in recent years.

Traditional methods for radar-based precipitation nowcasting rely on statistical models and physical assumptions (del Moral et al., 2018; Woo & Wong, 2017). Although these methods have the advantage of computational efficiency and high explainability, the chaotic and nonlinear nature of short-term precipitation means that various physical and statistical assumptions introduced in traditional methods have inherent limitations. These methods are only suitable for cases with smooth and simple motion patterns over short periods. Therefore, researchers explore the use of deep learning to mine the spatiotemporal correlations in precipitation nowcasting. These methods treat precipitation as a task of spatiotemporal prediction, predicting future radar echoes given the sequence of historical observations. By designing modules to better model the spatiotemporal dynamics in precipitation nowcasting, many attempts have provided solid improvements in the evaluation of Critical Success

---

\*Equal Contribution
†Corresponding Authors: Ben Fei (benfei@cuhk.edu.hk) and Lei Bai (bailei@pjlab.org.cn)

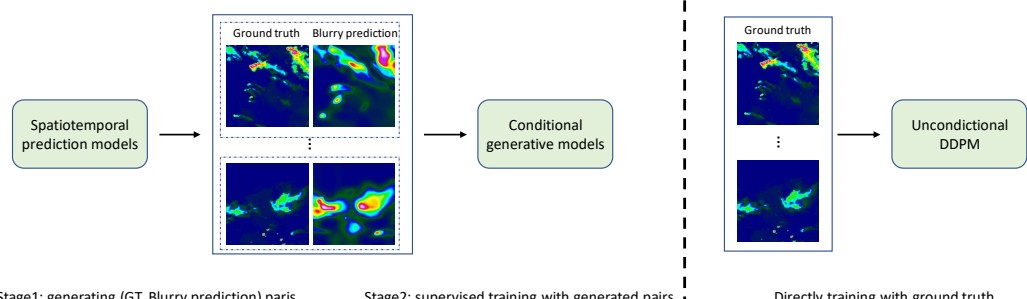

Figure 1: **Left**: Previous methods require two stages to generate predictions with local weather patterns, which generate (GT, blurry prediction) pairs in stage 1 and apply these pairs to supervise the training of conditional generative models in stage 2. **Right**: We propose to directly train an unconditional DDPM to convert blurry predictions into the distribution of ground truths.

Index (CSI) (Shi et al., 2015; Wang et al., 2022; Gao et al., 2022b;a). However, they often suffer from severe blur when the lead time of predictions increases (Gong et al.). Such blur hinders the predictions from containing local patterns that represent small-scale weather systems which are usually correlated to extreme precipitation events (Ravuri et al., 2021). Over the past 50 years, these extreme precipitations have caused 1.01 million related deaths, and over US$ 2.84 trillion economic losses. It results in continuous efforts to mitigate the blur in long-term predictions (Douris & Kim, 2021).

To avoid blurry predictions, researchers have proposed to utilize probabilistic methods to generate future radar echoes. Probabilistic methods, such as generative adversarial networks (GAN) or diffusion models (DM), sample different latent variables to express the stochasticity of chaotic future weather systems, enabling them to capture local weather patterns instead of the smooth mean value predicted in deterministic methods mentioned above (Ravuri et al., 2021; Gao et al., 2024; Zhao et al., 2024b). Furthermore, to simultaneously take advantage of probabilistic and deterministic modeling, recent methods utilize blurry predictions to capture the global movement of precipitation cloud clusters and harness probabilistic components to predict small-scale systems (Gong et al.; Yu et al., 2024; Zhang et al., 2023). However, these deterministic and probabilistic coupling methods also have several challenges. These methods train separate probabilistic models for different datasets, different deterministic models, and different lead times, which hampers the generalization capabilities of models. For instance, these methods require retraining when transferred from Shanghai to Hong Kong, as the deterministic models and observations applied by the local Meteorological Bureaus are different. Second, the training process of the probabilistic part of the coupling methods is complex. To train the probabilistic component, the blurry predictions and the corresponding ground truth are required to be provided in advance (Gong et al.; Yu et al., 2024; Zhang et al., 2023). As a result, there is an additional stage to prepare blurry predictions for the probabilistic model as shown in Figure 1. In summary, the generalization and flexibility of the deterministic and probabilistic coupling methods is limited.

Instead of capturing local patterns by the coupling training methods, we propose to rethink the blurry predictions from a direct perspective. The blurry predictions could be recognized as the results of blur kernels acting on the predictions with the distribution of real-world data. As shown in Appendix A.2, the blur kernels $\mathcal{K}_{S,T,M}$ are related to sample $S$, lead time of predictions $T$ and deterministic model $M$. Thus, we could obtain predictions without blurriness by solving the inverse of blur kernel $\mathcal{K}_{S,T,M}$. This perspective can lead to a totally different training paradigm to recover local weather patterns.

Motivated by the idea of deblurring, we propose our PostCast. The blur kernel $\mathcal{K}_{S,T,M}$ can be obtained by unsupervised estimation, which alleviates the burden of generating blurry predictions by complex spatiotemporal modeling. Besides, the process of obtaining the inverse solution of blur kernel $\mathcal{K}_{S,T,M}$ is generalizable, enabling our method to be flexibly applied in various datasets, deterministic models, and time steps. Specifically, our PostCast is a unified framework that integrates the generative prior inherent in the pre-trained diffusion model with zero-shot blur kernel estimation mechanism and auto-scale denoise guidance strategy to tackle blurry predictions across various datasets, models, and prediction lengths. Firstly, we utilize the pre-trained unconditional diffusion model on ImageNet from (Nichol & Dhariwal, 2021) for better initialization and fine-tune this dif-

fusion model on five precipitation datasets to enrich it with generative prior that can be utilized to generate high-quality precipitations. After the unconditional diffusion model is finetuned, we can utilize blurry predictions to guide the sampling process. In every sampling step, the diffusion model first generates a clean precipitation image $\tilde{x}_0$ from the noisy precipitation image $x_t$ by estimating the noise in $x_t$. We can add guidance with blurry predictions given by spatiotemporal prediction models on this intermediate variable $\tilde{x}_0$ to control the sampling process of the diffusion model. Since blurry prediction undergoes unknown degradation, a zero-shot blur kernel estimation mechanism and an auto-scale denoise guidance strategy are formulated to adaptively simulate this unknown degradation by kernel $\mathcal{K}_{S,T,M}$ at any blur modes. The parameters of the optimizable blur kernel are randomly initialized and optimized by the gradient of the distance metric between blurry prediction and the intermediate variable $\tilde{x}_0$ after the optimizable blur kernel. In this way, clean precipitation predictions guided by blurry predictions will be obtained after the sampling process of the diffusion model. Additionally, our method could also obtain the blur kernels that convert clean precipitation predictions into blurry ones, demonstrating the effectiveness of our optimized blur kernel. We demonstrate that our PostCast enhances the blurry predictions of existing methods on several precipitation datasets. Moreover, our PostCast can be adapted to a wide range of sample $S$, lead time of predictions $T$, and deterministic model $M$.

## 2 RELATED WORK

### 2.1 PRECIPITATION NOWCASTING

Notable progress has been achieved by applying deep learning in precipitation nowcasting (Chen et al., 2023; Han et al., 2024b; Xu et al., 2024; Han et al., 2024a; Zhao et al., 2024a). The initial attempts are deterministic methods focusing on spatiotemporal modeling. Researchers explore different spatiotemporal modeling structures such as RNN (Shi et al., 2015; Wang et al., 2022), CNN (Gao et al., 2022a; Tan et al., 2023), and Transformers (Gao et al., 2022b; Liu et al., 2020). However, these methods have a shortage of blurry predictions, which hamper the nowcasting of extreme events. Probabilistic methods are proposed to alleviate blurriness (Ravuri et al., 2021; Gao et al., 2024; Zhao et al., 2024b; Ji et al., 2022; 2023; Luo et al., 2022). DGMR (Ravuri et al., 2021) applies GAN to produce realistic and spatio-temporally consistent predictions to reduce blurriness. CLGAN (Ji et al., 2022; 2023) introduces an adversarial network with a novel long short-term memory to improve the nowcasting skills of heavy precipitation events. These GAN-based methods significantly increase the similarity between the predictions and ground truth, but the unstable training and inaccuracy of position and shape (Yu et al., 2024) hinder the further improvement on short-term forecasting. To further enhance precipitation nowcasting with accurate global movements, later methods combine blurry predictions with probabilistic models. Two-stage UA-GAN (Xu et al., 2022), DiffCast (Yu et al., 2024), and CasCast (Gong et al.) exploit how to generate small-scale weather pattern conditioning on the blurry predictions. Although these deterministic-probabilistic coupling methods achieve both global accuracy and local details, they are limited to blurry predictions in the training of probabilistic parts. From the perspective of deblur, our method is proposed to train the probabilistic part without the requirement of blurry predictions.

### 2.2 IMAGE DEBLUR WITH DIFFUSION MODELS

Diffusion-based models have been widely investigated in image deblurring tasks since it is capable of generating high-quality clean images (Song & Ermon, 2019; Ho et al., 2020; Song & Ermon, 2020; Fei et al., 2023). As a pioneering work, a U-Net architecture is trained in (Ho et al., 2020) with a denoising objective to iteratively refine the generated image starting from pure Gaussian noise. For instance, Austin et al. (2021) introduced Discrete Denoising Diffusion Probabilistic Models (D3PMs) as a way to generalize the multinomial diffusion model by incorporating non-uniform transition probabilities. Diffusion models can be conditioned on class labels or blurry images to further enhance the performance of deblurring effects (Dhariwal & Nichol, 2021; Saharia et al., 2022a). Ren et al. (2023) proposes the icDPM, which can better understand the blur and recover the clean image with the blurry input and guidance from the latent space of a regression network. However, these methods merely use the blurry image as a form of guidance, rather than attempting to simulate the blurriness itself, which renders them incapable of achieving a more precise and complete removal of the blur. Our model attempts to simulate and update the blur kernel and guidance scale in

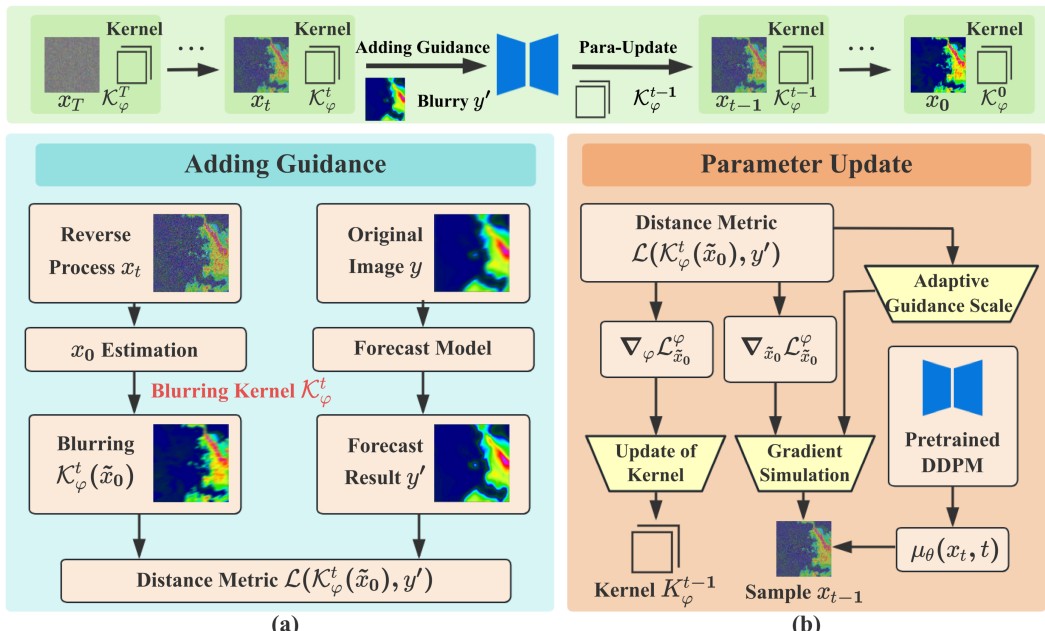

Figure 2: **Overview of PostCast for diffusion-based precipitation predictions deblurring. (a)** Unconditional diffusion model trained on 5 datasets is used to eliminate noise and estimate $\tilde{x}_0$ at every reverse step $t$, while an optimizable blur kernel is utilized to simulate the blur contained in the blurry image $y'$. PostCast introduces a distance metric at each step of the reverse process to quantify the loss between the blurry image $y'$ and the generated image $\tilde{x}_0$ after the blur kernel. **(b)** The Sampling process integrates a pre-trained diffusion model with guidance from the distance function. The gradient could be employed for updating and simulating a more accurate blur kernel.

real-time by introducing the guidance of blurry predicted images to form an unconditional diffusion model. The simulation of blur makes our model suitable for the general dataset while generating clean precipitation images with rich and accurate details.

## 3 METHOD

In precipitation nowcasting, the increasing blurriness with lead time is a crucial problem to be solved, as the blurriness impedes the accurate spatiotemporal modeling of small-scale weather patterns, which are related to most extreme precipitation events. Previous methods, utilizing the pairs of blurry predictions and observations to train models for the predictions of local weather patterns, are challenged to generalize well to blur modes that do not appear in training. Instead of directly predicting small-scale weather patterns by conditioning on historical observations and blurry predictions, we propose a new pipeline composed of estimating the blurriness in precipitation nowcasting directly and deblurring the blurry predictions with an unconditional diffusion model.

### 3.1 EXPLICITLY MODELING OF THE BLURRINESS IN PRECIPITATION NOWCASTING

There are many blur modes in blurry predictions of precipitation. On the one hand, the blurriness in predictions varies depending on the changes in lead time or fluctuations in weather conditions, influencing both the future probabilities and magnitudes of future changes. On the other hand, demonstrated by the visualizations in Appendix A.2, the differences in spatiotemporal modeling also have impacts on blurriness.

We first propose to explicitly model these blur modes in precipitation nowcasting with a unified formulation:

$$y' = conv(\mathcal{K}_{S,T,M}, y). \tag{1}$$

The above equation means that the blurry prediction $y'$ is recognized as implementing a convolution operation on the prediction $y$ with local weather patterns similar to observations. $\mathcal{K}_{S,T,M}$, represented by a $n \times n$ learnable matrix, is the kernel of convolution. The parameters of $\mathcal{K}_{S,T,M}$ vary

according to sample ($S$), lead time ($T$), and prediction model ($M$), as blur modes are influenced by weather conditions, lead time, and spatiotemporal modeling.

### 3.2 Unsupervised deblur for any blur modes in precipitation nowcasting

Inspired by the formulation of Equation 1, fuzzy prediction could be tackled by solving the fuzzy inverse problem $\mathcal{K}_{S,T,M}^{-1}$. However, in precipitation nowcasting, weather conditions vary with space and time, lead time changes in different application scenarios, and spatiotemporal modeling continuously advances. As a result, it is prohibited to generalize to all blur modes in precipitation nowcasting by supervised training with pairs composed of blurry predictions and observations.

To cope with countless blur modes in precipitation nowcasting, we proposed an unsupervised deblurring method based on a pre-trained unconditional diffusion model. Specifically, there is a **zero-shot blur estimation mechanism** and an **auto-scale gradient guidance strategy** to generalize our method to any blur modes in precipitation nowcasting.

#### 3.2.1 Zero-shot blur estimation mechanism

As shown in Figure 2, our method adds guidance with the blur kernel $\mathcal{K}_{S,T,M}$ and blurry prediction $y'$ in each reverse step of the pre-trained diffusion model. The parameter of $\mathcal{K}_{S,T,M}$ is randomly initialized and dynamically optimized at each step of the sampling process. In each reverse steps, there are two parts named "Adding Guidance" and "Parameter Update", respectively.

**Adding Guidance.** As shown in Figure 2(**a**), during this process, the generated radar image $\tilde{x}_0$ from pre-trained DDPM, which is calculated by estimating and eliminating the noise contained in $x_t$, undergoes convolution function with blur kernel $\mathcal{K}_\varphi^t$ [1] to establish a distance metric $\mathcal{L} = \mathcal{L}(\mathcal{K}_\varphi^t(\tilde{x}_0), y')$ with blurry prediction $y'$. Detailed introduction of $\tilde{x}_0$ can be found in the **Appendix A.4**. Distance function quantifies the discrepancy between deblurred maps convolved with the blurry kernel and the blurry prediction map, aiming to generate outputs that are closer to $y'$ after being subjected to the simulated blurry effect. And the guidance from the blurry prediction ensures the accuracy of the model's deblurring process through "Parameter Update", while the blur kernel connects the blurry prediction $y'$ and the generated radar image $\tilde{x}_0$.

**Parameter Update.** As shown in Figure 2(**b**), after calculating the distance function for incorporating guidance from blurry maps, the mean of $x_t$ is updated to sample $x_{t-1}$, while concurrently updating the convolution kernel parameters to more accurately simulate blurry in subsequent steps. To implement our **zero-shot blur estimation mechanism**, we employ $\nabla_\varphi \mathcal{L}_{\varphi,\tilde{x}_0}$, the gradients of distance metric $\mathcal{L}$ respect to kernel parameter $\varphi_t$, to estimate the blur kernel from scratch by dynamically updating the parameter itself. Additionally, the distance metric $\mathcal{L}$ also provides the gradients respect to $x_t$, $\nabla_{\tilde{x}_0} \mathcal{L}_{\varphi,\tilde{x}_0}$, which is utilized to guide the sampling of $x_{t-1}$.

Specifically, the sampling process of the diffusion model transforms distribution $p_\theta(x_{t-1}|x_t)$ into conditional distribution $p_\theta(x_{t-1}|x_t, y')$. Previous work (Dhariwal & Nichol, 2021) have derived the conditional transformation formula in the reverse process:

$$\log p_\theta(x_t|x_{t+1}, y') = \log\left(p_\theta(x_t|x_{t+1})p(y'|x_t)\right) + N_1 \tag{2}$$

$$\approx \log p_\theta(z) + N_2 \quad z \sim \mathcal{N}(z; \mu_\theta(x_t, t) + \Sigma \nabla_{x_t} \log p(y'|x_t)|_{x_t=\mu}, \Sigma I), \tag{3}$$

where $N_1 = -\log p_\theta(y'|x_{t+1})$, $N_2$ is a constant related to the gradient term $\nabla_{x_t} \log p(y'|x_t)|_{x_t=\mu}$. And the variance of the reverse process $\Sigma = \Sigma_\theta(x_t)$ is set as a constant. Detailed derivation and proof are shown in the **Appendix A.4**. Based on this derivation, reverse process $p_\theta(x_{t-1}|x_t, y')$ integrates the gradient to update the mean $\mu_\theta(x_t, t)$ generated from the pretrained DDPM. We exploit the gradient of distance metric $\mathcal{L}$ to approximate the value of $\nabla_{x_t} \log p(y'|x_t)$:

$$\nabla_{x_t} \log p(y'|x_t)|_{x_t=\mu} = -s\nabla_{x_t}\mathcal{L}(\mathcal{K}_\varphi^t(\tilde{x}_0), y'). \tag{4}$$

#### 3.2.2 Auto-scale gradient guidance strategy

Among them, $s$ is the scaling factor employed to control the degree of guidance and plays a vital role in the quality of radar image generation. However, as there are numerous blur modes in precipitation

---

[1] $\mathcal{K}_\varphi^t$ represents the blur kernel $\mathcal{K}_{S,T,M}$ with parameter $\varphi$ at step $t$ in the reverse progress

---

**Algorithm 1** Guided diffusion model with the guidance of blurry prediction $y'$. An unconditional diffusion model $\epsilon_\theta(x_t, t)$ fine-tuned on 5 datasets is given.

---

**Input:** Blurry prediction $y'$, optimized blur kernel $\mathcal{K}$ with parameters $\varphi$, learning rate $l$, guidance scale $s$, distance metric $\mathcal{L}$.
**Output:** Deblurred prediction $x_0$ conditioned on $y'$. Sample $x_T$ from $\mathcal{N}(0, I)$

1: **for all** t from T to 1 **do**
2:     $\tilde{x}_0 = \frac{x_t}{\sqrt{\bar{\alpha}_t}} - \frac{\sqrt{1-\bar{\alpha}_t}\epsilon_\theta(x_t,t)}{\sqrt{\bar{\alpha}_t}}$
3:     $\mathcal{L}_{\varphi,\tilde{x}_0} = \mathcal{L}(y', \mathcal{K}_\varphi^t(\tilde{x}_0))$
4:     $s = -\frac{(x_t-\mu)^T g + C}{\mathcal{L}(\mathcal{K}_\varphi^t(\tilde{x}_0), y')}$
5:     $\tilde{x}_0 \leftarrow \tilde{x}_0 - \frac{s(1-\bar{\alpha}_t)}{\sqrt{\bar{\alpha}_{t-1}}\beta_t}\nabla_{\tilde{x}_0}\mathcal{L}_{\varphi,\tilde{x}_0}$
6:     $\tilde{\mu}_t = \frac{\sqrt{\bar{\alpha}_{t-1}}\beta_t}{1-\bar{\alpha}_t}\tilde{x}_0 + \frac{\sqrt{\bar{\alpha}_t}(1-\bar{\alpha}_{t-1})}{1-\bar{\alpha}_t}x_t$
7:     $\tilde{\beta}_t = \frac{1-\bar{\alpha}_{t-1}}{1-\bar{\alpha}_t}\beta_t$
8:     Sample $x_{t-1}$ from $\mathcal{N}(\tilde{\mu}_t, \tilde{\beta}_t I)$
9:     $\varphi \leftarrow \varphi - l\nabla_\varphi\mathcal{L}_{\varphi,\tilde{x}_0}$
10: **end for**
11: **return** $x_0$

---

nowcasting, it is difficult to set the guidance scale $s$ for each blurry mode. Instead, we propose an **auto-scale gradient guidance strategy** to adaptively derive $s$ for any blurry prediction from an empirical formula:

$$s = -\frac{(x_t-\mu)^T g + C}{\mathcal{L}(\mathcal{K}_\varphi^t(\tilde{x}_0), y')}, \tag{5}$$

where $g$ refers to the $\nabla_{x_t}\log p(y'|x_t)|_{x_t=\mu}$ and $C = \log p(y'|x_t)|_{x_t=\mu}$. The detailed derivation process of $s$ is shown in **Appendix A.4**.

The details of PostCast are shown in Algorithm 1. PostCast undergoes $T$ reverse steps to gradually restore pure Gaussian noise $x_T \sim \mathcal{N}(0, I)$ to high-quality precipitation images. For each reverse steps $t$, mean $\mu_\theta(x_t, t)$ is integrated with $\nabla_{x_t}\log p(y'|x_t)$ to sample $x_{t-1}$. The blur kernel parameter $\varphi$ which is related to reverse step $t$ is dynamically updated by the gradients of distance metric $\mathcal{L}_{\varphi,\tilde{x}_0}$. The optimizable blur kernel $\mathcal{K}_\varphi^t$ and auto-scale guidance factor $s$ enable the model to achieve blur simulation and flexibly eliminate blurriness for any blur modes in precipitation nowcasting.

## 4 EXPERIMENTS

This section includes the experiment setups and the analysis of the results. We begin with implementation details in Section 4.1, and evaluation metrics in Section 4.2. In Section 4.3, 4.4, and 4.5, we present comprehensive experiments exhibiting the high generalization ability of PostCast to enhance the extreme part of predictions generated by classical spatiotemporal methods. Finally, the ablation study of PostCast and further analysis of the blur kernel $\mathcal{K}_{S,T,M}$ and auto-scale guidance are presented in Section 4.6.

### 4.1 IMPLEMENTING DETAILS

We uniformly resize the radar images from all datasets to $256 \times 256$. Five datasets, including SEVIR (Veillette et al., 2020), HKO7 (Shi et al., 2017), TAASRAD19 (Franch et al., 2020), Shanghai (Chen et al., 2020), and SRAD2018 (SRAD, 2018), are selected to train the unconditional DDPM, while the other datasets (SCWDS CAP30 (Na et al., 2021), SCWDS CR (Na et al., 2021), MeteoNet (Larvor & Berthomier, 2021)) are prepared for out-of-distribution testing to evaluate the generalization of each method. More details of each dataset can be found in Appendix A.6. We follow (Dhariwal & Nichol, 2021) to train our DDPM. We utilize the pre-trained unconditional diffusion model on ImageNet for better initialization and fine-tune it on SEVIR, HKO7, TAARSARD19, Shanghai, and SRAD2018 using AdamW with $\beta_1 = 0.9$ and $\beta_2 = 0.999$. PostCast uses a blur kernel with a size of $9 \times 9$. To recover the prediction with a distribution of real observation, we implement our method with 1000-step DDPM. The cosine learning rate policy is used with initial

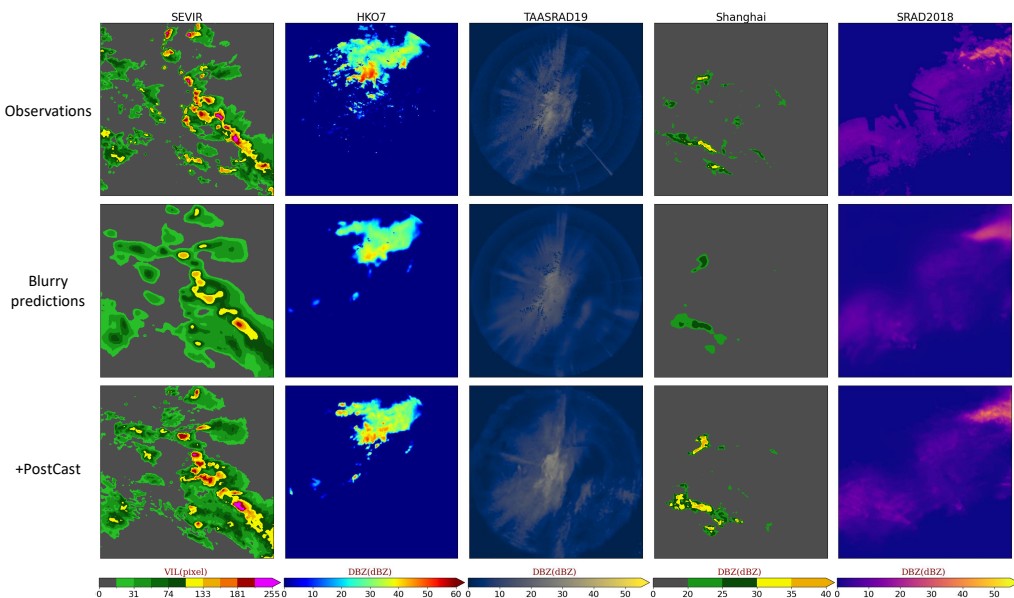

Figure 3: Visualization of applying our PostCast on 5 datasets at time step 12 when the spatiotemporal prediction model is TAU.

Table 1: The CSI scores of the highest thresholds evaluated by P1 (max pooling 1), P4 (max pooling 4), and P16 (max pooling 16) at time step 12 (about 1 hour lead time), as well as HSS and POD.

| Model | SEVIR | | | | | HKO7 | | | | | TAASRAD19 | | | | | Shanghai | | | | | SRAD2018 | | | | |
|---|---|---|---|---|---|---|---|---|---|---|---|---|---|---|---|---|---|---|---|---|---|---|---|---|---|
| | P1 | P4 | P16 | HSS↑ | POD↑ | P1 | P4 | P16 | HSS↑ | POD↑ | P1 | P4 | P16 | HSS↑ | POD↑ | P1 | P4 | P16 | HSS↑ | POD↑ | P1 | P4 | P16 | HSS↑ | POD↑ |
| TAU | 0.008 | 0.014 | 0.028 | 0.383 | 0.372 | 0.051 | 0.064 | 0.104 | 0.390 | 0.332 | 0.010 | 0.017 | 0.021 | 0.276 | 0.220 | 0.023 | 0.029 | 0.040 | 0.323 | 0.277 | 0.031 | 0.028 | 0.025 | 0.308 | 0.229 |
| +ours | 0.043 | 0.074 | 0.163 | 0.402 | 0.438 | 0.060 | 0.127 | 0.289 | 0.386 | 0.389 | 0.044 | 0.072 | 0.127 | 0.300 | 0.311 | 0.051 | 0.102 | 0.216 | 0.338 | 0.421 | 0.100 | 0.136 | 0.170 | 0.338 | 0.421 |
| PredRNN | 0.013 | 0.014 | 0.017 | 0.378 | 0.358 | 0.006 | 0.008 | 0.018 | 0.352 | 0.304 | 0.008 | 0.010 | 0.012 | 0.237 | 0.178 | 0.009 | 0.012 | 0.020 | 0.268 | 0.216 | 0.025 | 0.044 | 0.051 | 0.289 | 0.221 |
| +ours | 0.059 | 0.083 | 0.161 | 0.397 | 0.432 | 0.050 | 0.110 | 0.266 | 0.350 | 0.371 | 0.038 | 0.064 | 0.138 | 0.279 | 0.275 | 0.031 | 0.069 | 0.167 | 0.303 | 0.349 | 0.086 | 0.139 | 0.256 | 0.303 | 0.349 |
| SimVP | 0.015 | 0.016 | 0.024 | 0.389 | 0.385 | 0.042 | 0.049 | 0.067 | 0.409 | 0.358 | 0.000 | 0.000 | 0.002 | 0.242 | 0.181 | 0.025 | 0.030 | 0.060 | 0.303 | 0.258 | 0.037 | 0.049 | 0.047 | 0.331 | 0.258 |
| +ours | 0.045 | 0.069 | 0.140 | 0.409 | 0.462 | 0.054 | 0.116 | 0.264 | 0.385 | 0.409 | 0.021 | 0.035 | 0.051 | 0.298 | 0.275 | 0.044 | 0.094 | 0.212 | 0.303 | 0.390 | 0.095 | 0.172 | 0.272 | 0.303 | 0.390 |
| EarthFormer | 0.032 | 0.024 | 0.023 | 0.374 | 0.357 | 0.025 | 0.025 | 0.035 | 0.390 | 0.334 | 0.019 | 0.021 | 0.028 | 0.266 | 0.204 | 0.021 | 0.029 | 0.055 | 0.304 | 0.253 | 0.036 | 0.034 | 0.040 | 0.311 | 0.244 |
| +ours | 0.045 | 0.070 | 0.131 | 0.403 | 0.427 | 0.066 | 0.125 | 0.257 | 0.392 | 0.395 | 0.044 | 0.067 | 0.143 | 0.286 | 0.283 | 0.048 | 0.098 | 0.226 | 0.321 | 0.396 | 0.095 | 0.155 | 0.276 | 0.321 | 0.396 |
| DiffCast | 0.049 | 0.070 | 0.186 | 0.362 | 0.378 | 0.061 | 0.113 | 0.255 | 0.385 | 0.375 | 0.044 | 0.076 | 0.174 | 0.267 | 0.260 | 0.050 | 0.097 | 0.218 | 0.309 | 0.282 | 0.071 | 0.124 | 0.257 | 0.313 | 0.307 |
| CasCast | 0.039 | 0.067 | 0.156 | 0.335 | 0.422 | 0.054 | 0.108 | 0.235 | 0.343 | 0.454 | 0.040 | 0.064 | 0.128 | 0.221 | 0.301 | 0.034 | 0.068 | 0.167 | 0.259 | 0.336 | 0.061 | 0.109 | 0.240 | 0.269 | 0.333 |
| DGMR | 0.003 | 0.010 | 0.062 | 0.122 | 0.235 | 0.018 | 0.055 | 0.210 | 0.210 | 0.182 | 0.015 | 0.038 | 0.120 | 0.097 | 0.091 | 0.007 | 0.028 | 0.132 | 0.105 | 0.103 | 0.022 | 0.066 | 0.213 | 0.114 | 0.147 |
| STRPM | 0.007 | 0.023 | 0.060 | 0.307 | 0.296 | 0.010 | 0.027 | 0.078 | 0.263 | 0.196 | 0.005 | 0.016 | 0.054 | 0.186 | 0.138 | 0.121 | 0.428 | 0.128 | 0.236 | 0.201 | 0.034 | 0.076 | 0.171 | 0.251 | 0.197 |
| DGP | 0.020 | 0.042 | 0.070 | 0.372 | 0.355 | 0.039 | 0.083 | 0.187 | 0.372 | 0.328 | 0.018 | 0.041 | 0.094 | 0.238 | 0.196 | 0.029 | 0.070 | 0.160 | 0.282 | 0.271 | 0.044 | 0.089 | 0.176 | 0.298 | 0.239 |

learning rates 0.0002 for PostCast and the $\beta_t$ we utilize undergoes a linear increase from $\beta_1 = 10^{-4}$ to $\beta_T = 0.02$.

## 4.2 EVALUATION METRIC

We choose the Critical Success Index, Probability of Detection (POD), and Heidke Skill Score (HSS) for evaluation. For each dataset, the thresholds with the highest intensity are selected to quantitatively evaluate the accuracy of predictions for extreme events by CSI. The blurriness in deterministic predictions influences the modeling of small-scale patterns, which are usually correlated to extreme precipitation events. Meanwhile, average CSI, POD, and HSS scores across different thresholds are evaluated to provide a more comprehensive evaluation of precipitation with different intensities. Details of evaluated threshold $\tau$ are given in **Appendix. A.5**. Before calculating these scores, we set the predicted and observed pixel values less than $\tau$ to 0 otherwise 1. These binary values enable us to determine the true positive (TP), false negative (FN), true negative (TN), and false positive (FP) counts. The formulations are: $CSI = \frac{TP}{TP+FN+FP}$. $POD = \frac{TP}{TP+FN}$. $HSS = \frac{2 \times (TP \times TN - FN \times FP)}{(TP+FN) \times (FN+TN) + (TP+FP) \times (FP+TN)}$. CSI and POD values vary between 0 and 1, while HSS value varies between -1 and 1, with values approaching 1 indicating a higher level of agreement between the predicted and observed results.

## 4.3 EVALUATION ON MULTIPLE DATASETS

Table 1 presents the quantitive evaluation results of our method's performance gain for extreme precipitation nowcasting and general predictions. Specifically, TAU (Tan et al., 2023), PredRNN (Wang et al., 2022), SimVP (Gao et al., 2022a), and EarthFormer (Gao et al., 2022b) are all independently trained on the evaluated datasets. We compare our method with four supervised GAN- or

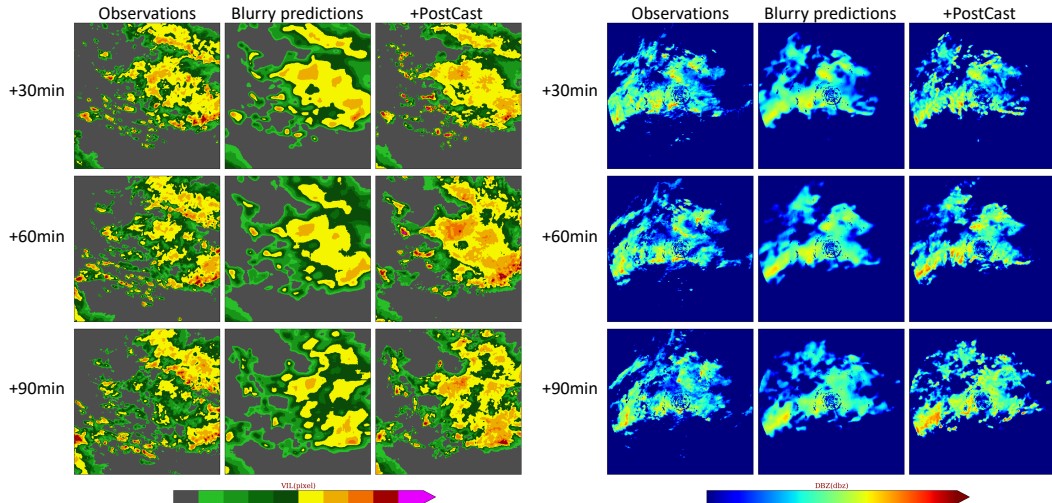

Figure 4: **Left**: Visualizations on SEVIR. **Right**: Visualizations on HKO7. Both blurry predictions are given by model TAU. The lead times of predictions are 30 minutes, 60 minutes, and 90 minutes.

Table 2: The CSI scores of the highest thresholds evaluated by P16 at different lead times.

| Model | HKO7 | | | SEVIR | | | Model | HKO7 | | | SEVIR | | |
|---|---|---|---|---|---|---|---|---|---|---|---|---|---|
| | 30min | 60min | 90min | 30min | 60min | 90min | | 30min | 60min | 90min | 30min | 60min | 90min |
| TAU | 0.216 | 0.104 | 0.082 | 0.029 | 0.028 | 0.013 | SimVP | 0.183 | 0.067 | 0.083 | 0.083 | 0.024 | 0.025 |
| +ours | 0.369 | 0.326 | 0.256 | 0.228 | 0.144 | 0.098 | +ours | 0.394 | 0.313 | 0.266 | 0.252 | 0.116 | 0.093 |
| PredRNN | 0.127 | 0.018 | 0.021 | 0.079 | 0.017 | 0.025 | EarthFormer | 0.143 | 0.035 | 0.042 | 0.079 | 0.023 | 0.022 |
| +ours | 0.349 | 0.190 | 0.216 | 0.227 | 0.104 | 0.086 | +ours | 0.376 | 0.266 | 0.255 | 0.250 | 0.128 | 0.106 |

Diffusion-based methods which are jointly trained on these five datasets (DiffCast (Yu et al., 2024), CasCast (Gong et al.), DGMR (Ravuri et al., 2021), STRPM (Chang et al., 2022)), and an unsupervised GAN-based method (DGP (Pan et al., 2021)). Our unconditional DDPM is also jointly on the five datasets in Table 1, which makes this evaluation in-domain. As shown in Table 1, our method can be applied to all of these prediction methods including RNN-based, CNN-based, and Transformer-based. It demonstrates that our method is not sensitive to the way how the spatiotemporal correlations are modeled. On each dataset, there are significant improvements in extreme precipitation evaluation when applying our method, which is attributed to the local weather patterns recovered by our method as exhibited in Figure 3. Besides, it reveals the potential of our method to adapt to different blur modes in precipitation nowcasting related to datasets. For general prediction quality evaluated by avg-HSS and avg-POD scores, it is significantly improved after applying our PostCast. In summary, PostCast demonstrates outstanding improvement across different prediction models and datasets, proving its effectiveness and flexibility in recovering local weather patterns and increasing the general precipitation nowcasting.

## 4.4 EVALUATION AT ANY LEAD TIME

Encouraged by the generality of PostCast among datasets and models, in this section, we exhibit the ability of PostCast to be generalized to arbitrary lead times such as 30 min, 60 min, and 90 min, within a zero-shot manner. An example is visualized in Figure 4, showing our method enhances local details and boosts the predictions of extreme values. As shown in Table 2, we conduct experiments on SEVIR and HKO7. For both datasets, no matter which spatiotemporal prediction models are used, our method consistently increases the CSI scores of the highest thresholds (32.24 $kg/m^2$ for SEVIR, and 30 $mm/h$ for HKO7). Specifically, in SEVIR, the highest CSI scores of 30 min, 60 min, and 90 min reach 0.252, 0.144, and 0.106, respectively. The highest CSI scores evaluated on HKO7 reach 0.394, 0.326, and 0.266 for the lead time of 30 min, 60 min, and 90 min. The consistent gain indicates the generality of our method among different lead times.

## 4.5 DEBLURRING ON OUT-OF-DISTRIBUTION DATASETS

We compare our PostCast with other methods described in Section 4.3 on three out-of-distribution datasets. The quantitative results are presented in Table 3. CAP30 and CR represent different modalities (constant altitude plan of 3 km and composite reflectivity) in SCWDS. These 3 datasets

Table 3: Evaluation on out-of-distribution datasets. The CSI scores are calculated within the highest thresholds at a lead time of 1 hour. P1, P4, and P16 indicate max pooling 1, max pooling 4, and max pooling 16, respectively.

| Model | SCWDS CAP30 | | | | | SCWDS CR | | | | | MeteoNet | | | | |
|---|---|---|---|---|---|---|---|---|---|---|---|---|---|---|---|
| | P1 | P4 | P16 | HSS | POD | P1 | P4 | P16 | HSS | POD | P1 | P4 | P16 | HSS | POD |
| TAU | 0.038 | 0.042 | 0.064 | 0.312 | 0.280 | 0.082 | 0.075 | 0.082 | 0.413 | 0.384 | 0.001 | 0.003 | 0.016 | 0.272 | 0.240 |
| +CasCast | 0.067 | 0.102 | 0.224 | 0.306 | 0.315 | 0.101 | 0.145 | 0.258 | 0.380 | 0.377 | **0.029** | **0.067** | 0.128 | 0.271 | 0.294 |
| +DiffCast | 0.023 | 0.050 | 0.166 | 0.157 | 0.235 | 0.051 | 0.101 | 0.245 | 0.232 | **0.554** | 0.006 | 0.015 | 0.063 | 0.079 | 0.065 |
| +ours | **0.075** | **0.126** | **0.269** | **0.345** | **0.428** | **0.143** | **0.214** | **0.338** | **0.444** | 0.549 | 0.024 | 0.059 | **0.182** | **0.288** | **0.344** |
| PredRNN | 0.003 | 0.004 | 0.008 | 0.239 | 0.203 | 0.040 | 0.043 | 0.066 | 0.351 | 0.323 | 0.000 | 0.000 | 0.002 | 0.230 | 0.190 |
| +CasCast | 0.035 | 0.056 | 0.129 | 0.252 | 0.231 | 0.086 | 0.139 | 0.283 | 0.337 | 0.390 | 0.010 | 0.030 | 0.101 | 0.249 | 0.238 |
| +DiffCast | 0.017 | 0.035 | 0.102 | 0.157 | 0.235 | 0.066 | 0.105 | 0.230 | 0.349 | 0.341 | 0.006 | 0.019 | 0.076 | 0.241 | 0.215 |
| +ours | **0.060** | **0.126** | **0.267** | **0.331** | **0.351** | **0.140** | **0.206** | **0.315** | **0.405** | **0.485** | 0.022 | 0.050 | 0.148 | 0.283 | 0.298 |
| SimVP | 0.025 | 0.026 | 0.035 | 0.312 | 0.276 | 0.056 | 0.046 | 0.041 | 0.410 | 0.373 | 0.000 | 0.000 | 0.002 | 0.281 | 0.245 |
| +CasCast | 0.069 | 0.111 | 0.226 | 0.314 | 0.319 | 0.098 | 0.134 | 0.242 | 0.382 | 0.364 | 0.030 | 0.053 | **0.149** | 0.282 | 0.305 |
| +DiffCast | 0.024 | 0.044 | 0.129 | 0.224 | 0.201 | 0.047 | 0.071 | 0.169 | 0.295 | 0.280 | 0.017 | 0.037 | 0.105 | 0.240 | 0.216 |
| +ours | **0.085** | **0.136** | **0.255** | **0.367** | **0.436** | **0.140** | **0.205** | **0.296** | **0.459** | **0.545** | 0.025 | 0.054 | 0.147 | **0.316** | **0.391** |
| EarthFormer | 0.021 | 0.024 | 0.036 | 0.298 | 0.258 | 0.072 | 0.065 | 0.063 | 0.417 | 0.406 | 0.000 | 0.003 | 0.008 | 0.259 | 0.219 |
| +CasCast | 0.050 | 0.089 | 0.190 | 0.296 | 0.287 | 0.100 | 0.130 | 0.223 | 0.381 | 0.383 | **0.019** | 0.055 | 0.159 | 0.266 | 0.265 |
| +DiffCast | 0.041 | 0.071 | 0.175 | 0.299 | 0.278 | 0.101 | 0.144 | 0.268 | 0.407 | 0.417 | 0.009 | 0.029 | 0.096 | 0.263 | 0.243 |
| +ours | **0.070** | **0.117** | **0.241** | **0.350** | **0.404** | **0.141** | **0.211** | **0.326** | **0.444** | **0.570** | **0.019** | **0.058** | **0.164** | **0.287** | **0.320** |
| DGMR | 0.018 | 0.048 | 0.160 | 0.161 | 0.153 | 0.039 | 0.090 | 0.240 | 0.207 | 0.208 | 0.019 | 0.057 | 0.192 | 0.123 | 0.131 |
| STRPM | 0.014 | 0.049 | 0.160 | 0.234 | 0.197 | 0.029 | 0.080 | 0.201 | 0.296 | 0.264 | 0.014 | 0.046 | 0.145 | 0.192 | 0.155 |
| DGP | 0.027 | 0.059 | 0.111 | 0.294 | 0.258 | 0.071 | 0.083 | 0.099 | 0.409 | 0.397 | 0.037 | 0.082 | 0.186 | 0.250 | 0.218 |

Table 4: The ablation study on the optimizable convolutional kernel and the adaptive guidance scale. The CSI scores with the highest thresholds are calculated by P1 (max pooling 1), P4 (max pooling 4), and P16 (max pooling 16). The lead time is 60min for SEVIR and 72min for HKO7 (both 12 steps). In cells marked with a "✓", the corresponding module is employed, while cells with "✗" indicate a fixed guidance scale of 3500 or a random blur kernel with a mean value of 0.6, which is consistent with the initial value setting of PostCast, and remain unchanged throughout each step of the reverse process. The reason for these settings can be found in Appendix A.7

| Methods | Dynamic Update | | SEVIR | | | HKO7 | | |
|---|---|---|---|---|---|---|---|---|
| | Kernel | Guidance Scale | P1 | P4 | P16 | P1 | P4 | P16 |
| Model A | ✗ | ✗ | 0.010 | 0.019 | 0.048 | 0.030 | 0.088 | 0.219 |
| Model C | ✓ | ✗ | 0.038 | 0.064 | 0.115 | 0.059 | 0.102 | 0.232 |
| **PostCast** | ✓ | ✓ | 0.045 | 0.070 | 0.131 | 0.066 | 0.125 | 0.257 |

are excluded from both the fine-tuning of our unconditional DDPM. Our method notably improves the nowcasting of extreme precipitation and general precipitation nowcasting even in the out-of-distribution datasets. Specifically, on SCWDS CAP30, the highest P16, HSS, and POD of our method reaches 0.269, 0.367, and 0.436, respectively, while other methods only achieve 0.226, 0.314, and 0.319. On MeteoNet, PostCast fulfills the best HSS and POD across all methods. Moreover, our method achieves competitive performance when applied on SimVP and EarthFormer, while on TAU and PredRNN, our method remarkably enhances the CSI scores, reaching 0.182 on TAU and 0.148 on PredRNN. Such gap in performance on different spatiotemporal prediction models further reveals the requirement for generalization to various spatiotemporal modeling approaches. To summarize, our method impressively enhances the blurry predictions even on out-of-distribution datasets.

## 4.6 ABLATION STUDY

In this section, we conduct ablation to validate the effectiveness of our proposed **zero-shot blur estimation mechanism** and **auto-scale gradient guidance strategy**. Additionally, further analysis of the blur kernel estimation and the auto-scale guidance is conducted.

Table 4 presents the results of the ablation study on SEVIR and HKO7. "Kernel" stands for the **zero-shot blur estimation mechanism**, and "Guidance Scale" represents the **auto-scale gradient guidance strategy**. As shown in Table 4, when the deblur progress is equipped with neither "Kernel" nor "Guidance Scale", it exhibits relatively low CSI scores for extreme precipitation nowcasting. When "Kernel" is solely applied, there are significant gains on both SEVIR and HKO7, especially on the CSI-P1 evaluated pixel-wisely. In particular, the CSI-P1 of SEVIR reached 0.038 and that of HKO7 reached 0.059, indicating the importance of estimating the blur modes. Further, "Guidance Scale" improves the gain of "Kernel", which suggests adaptively scaling the guidance contributes to better guidance.

Further experiments are conducted to reveal the influence of "Kernel" and "Guidance Scale". As shown in Figure 5 (a), for the SEVIR dataset, the mean of the kernel stabilizes around 2.65 at reverse

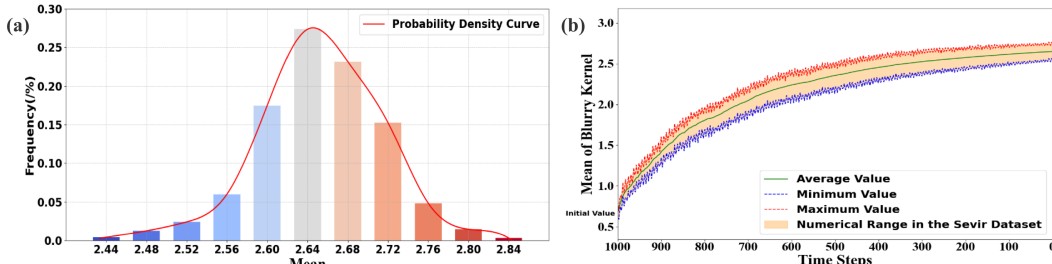

Figure 5: **(a)** The distribution of the mean of blur kernel at reverse step $t = 0$ on SEVIR dataset. **(b)** The variation in the mean of the blur kernel with reverse steps on SEVIR dataset.

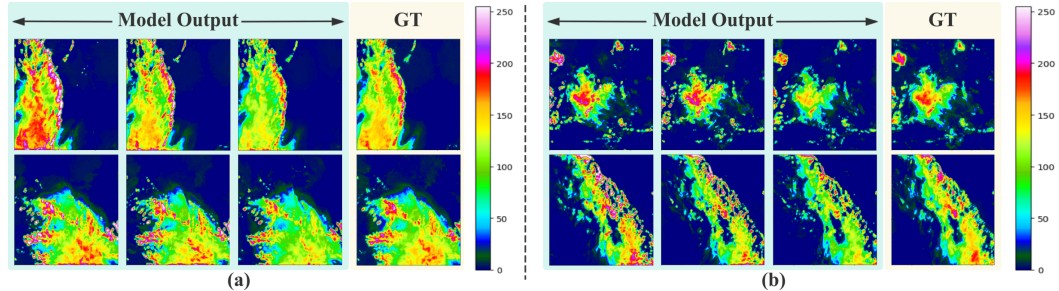

Figure 6: **(a)** Variations in the guidance scale can result in different intensities of precipitation. From left to right, the guidance scale values are 1.25, 1, and 0.75 times our **auto-scale gradient guidance strategy**; **(b)** Different initial values of the blur kernel parameters affect the resultant precipitation intensity maps to varying degrees. As the initial values of parameters decrease from left to right, the precipitation intensities correspondingly diminish. The intermediate map in the model output represents the model's standard parameter settings, and the results are closest to the ground truth.

step $t = 0$. And for a single map from the SEVIR datasets, as illustrated in Figure 5 (b), the mean of optimizable blur kernel parameters increases with the sampling process. This increase in magnitude is influenced by the gradient of the distance metric with respect to the parameter. Ultimately, the mean value of convolutional kernel parameters gradually converges to approximately 2.65.

The visualization of Figure 10 indicates a close resemblance between the PostCast output map convolved with the optimizable blur kernel at time $t = 0$ and the prediction map, suggesting that the blur kernel effectively estimates the blur present within the prediction map, allowing the model to generate high-quality outcomes with faithfulness details that are similar to the ground truth.

Furthermore, the "Kernel" and "Guidance Scale" are capable of controlling the intensity of generated precipitation predictions through parameter adjustments. As illustrated in 6, **the zero-shot blur estimation mechanism**, combined with **auto-scale gradient guidance strategy**, enables the model to produce results that most closely approximate the ground truth in terms of small-scale structures and precipitation intensity. By manually increasing or decreasing the values of the guidance scale and blur kernel, we can either enhance or diminish the precipitation intensity in the generated images. This outcome demonstrates the model's substantial controllability, allowing it to meet diverse usage requirements with greater flexibility.

## 5 CONCLUSION

In this paper, we propose PostCast, a generalizable postprocessing method for precipitation nowcasting to enhance the local weather patterns and extreme precipitation nowcasting. Specifically, it integrates the generative prior in the pre-trained diffusion model with zero-shot blur kernel estimation and auto-scale denoise guidance to enhance blurry predictions. Experiments demonstrate that our method could increase the ability of extreme nowcasting for varying datasets, different lead times, and multiple spatiotemporal prediction models.

ACKNOWLEDGEMENTS

This work is Supported by Shanghai Artificial Intelligence Laboratory. This work was done during Junchao Gong's internship at Shanghai Artificial Intelligence Laboratory.

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

# A APPENDIX

## A.1 LIMITATIONS AND FUTURE WORK

Our work has a certain limitation: As a postprocessing method, the effectiveness of PostCast partially depends on the spatiotemporal prediction models. This may lead to poor performance if the blurry predictions have relatively low accuracy. Motivated by the importance of an accurate prediction, in the future, we will make an effort to improve our method with priors of spatiotemporal prediction modeling.

## A.2 VARING BLUR MODES

In this section, we visualize the blurry predictions with different blur modes. As shown in Figure 7, TAU has a sharper prediction than other models, which indicates the blur modes vary with the spatiotemporal modeling approaches. We also demonstrate that the blur modes change with the lead time in Figure 8. Longer lead time results in a stronger blurry mode and weaker extreme predictions. Figure 9 presents the different blur modes across datasets or samples. In this case, the samples of

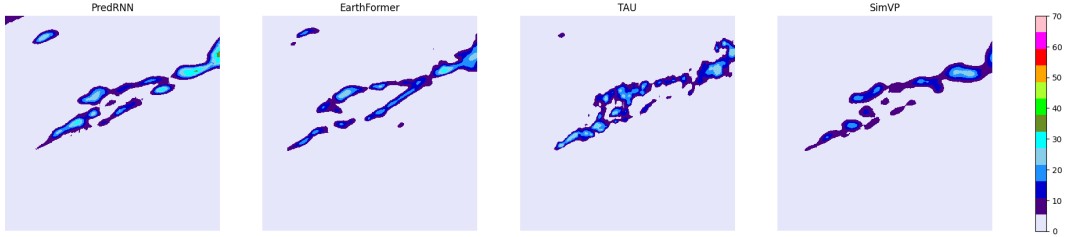

Figure 7: Visualization of predictions of different spatiotemporal prediction models at the lead time of 1 hour on MeteoNet.

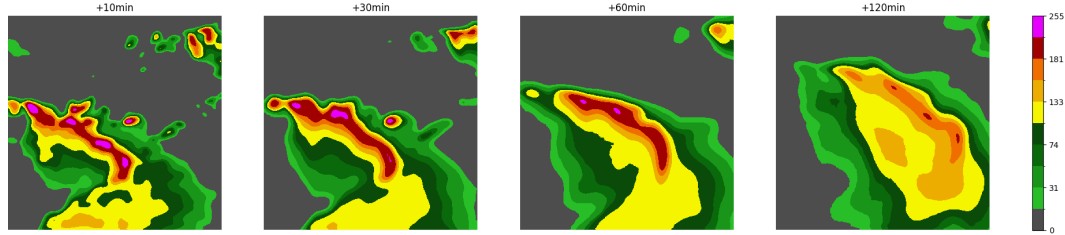

Figure 8: Visualization of predictions of EarthFormer on SEVIR at different lead times.

SEVIR and TAASRAD19 suffer from a severe blurry mode compared, while those of MeteoNet and HKO7 have more local weather patterns. In summary, the blur modes vary with spatiotemporal modeling approaches, lead times, and weather conditions which are represented by datasets and samples. There is an urgent requirement for a method to deal with the countless blur modes.

## A.3 PRELIMINARY

Diffusion model is a generative framework that encompasses both the forward and reverse processes. The forward process gradually destroys the original training data $x_0$ by adding Gaussian noise successively, which is defined as a Markov chain:

$$q(x_1, \cdots, x_T | x_0) = \prod_{t=1}^{T} q(x_t | x_{t-1}). \tag{6}$$

Pure Gaussian noise can be obtained after $T$ diffusion steps when $T$ is large enough. Each diffusion steps is defined by the given parameter series $\beta_t$, which refers to the variance of the forward process. It can be set as a known constant or learned with a separate neural network head (Nichol & Dhariwal, 2021).

$$q(x_t | x_{t-1}) = \mathcal{N}(x_t; \sqrt{1 - \beta_t} x_{t-1}, \beta_t I). \tag{7}$$

The reverse process is the inversion of the forward process, aiming to simulate noise from the noise distribution at each diffusion step and recover data from it. However, the mean and variance of the reverse process conditional distribution $p_\theta(x_{t-1} | x_t) = \mathcal{N}(x_{t-1}; \mu(x_t, t), \Sigma I)$ is difficult to calculate directly. Therefore, we need to learn a noise simulation function $\epsilon_\theta(x_t, t)$ parameterized by parameter $\theta$ to approximate the mean of the conditional probabilities, which enables the model to simulate and eliminate noise in the data sampled from the reverse process.

Bayesian formula can be utilized to transform the conditional distribution as follows: Bayesian formulas can be employed to derive the mean and variance for each step in the reverse process $p_\theta(x_{t-1} | x_t)$.

$$q(x_{t-1} | x_t, x_0) = q(x_t | x_{t-1}, x_0) \frac{q(x_{t-1} | x_0)}{q(x_t | x_0)}. \tag{8}$$

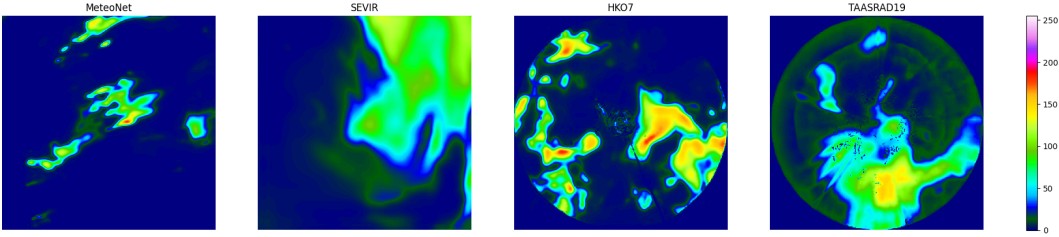

Figure 9: Visualization of 1 hour's prediction of EarthFormer on different datasets.

Directly expanding the three terms at the right-hand of Equation 8, the mean $\mu_\theta$ and variance $\Sigma_\theta$ can be represented by the following equation:

$$\mu_\theta(x_t, t) = \frac{1}{\sqrt{\alpha_t}}(x_t - \frac{\beta_t}{\sqrt{1-\bar{\alpha}_t}}\epsilon_\theta(x_t, t)) \tag{9}$$

$$\Sigma_\theta(x_t) = \frac{1-\bar{\alpha}_{t-1}}{1-\bar{\alpha}_t}\beta_t \tag{10}$$

where $\alpha_t = 1 - \beta_t$ and $\bar{\alpha}_t = \prod_{i=1}^{t}\alpha_t$, which indicates that the variance of the reverse process $\Sigma = \Sigma_\theta(x_t)$ is a constant.

Diffusion model utilizes maximum likelihood estimation to obtain the probability distribution of Markov transition in the reverse process. Specifically, the noise prediction function $\epsilon_\theta(x_t, t)$ is trained with the purpose of optimizing the following surrogate denoising objective.

$$E_{\epsilon\sim\mathcal{N}(\prime,\mathcal{I}),t\sim[0,T]}[\|\epsilon - \epsilon_\theta(x_t, t)\|^2]. \tag{11}$$

### A.4 EMPIRICAL FORMULA OF GUIDANCE SCALE

In the reverse process of the diffusion model, we incorporated guidance from $y\prime$ to transform the initial reverse denoising distribution $p_\theta(x_t \mid x_{t+1})$ into a conditional distribution $p_\theta(x_t \mid x_{t+1}, y)$. Proven by (Dhariwal & Nichol, 2021), this distribution can be further simplified:

$$p_\theta(x_t \mid x_{t+1}, y) = \frac{p_\theta(x_t, x_{t+1}, y)}{p_\theta(x_{t+1}, y)} \tag{12}$$

$$= \frac{p_\theta(x_t, x_{t+1}, y)}{p_\theta(y \mid x_{t+1})p_\theta(x_{t+1})} \tag{13}$$

$$= \frac{p_\theta(x_t \mid x_{t+1})p_\theta(y \mid x_t, x_{t+1})p_\theta(x_{t+1})}{p_\theta(y \mid x_{t+1})p_\theta(x_{t+1})} \tag{14}$$

$$= \frac{p_\theta(y \mid x_t, x_{t+1})p_\theta(x_t \mid x_{t+1})}{p_\theta(y \mid x_{t+1})} \tag{15}$$

$$= \frac{p_\theta(y \mid x_t)p_\theta(x_t \mid x_{t+1})}{p_\theta(y \mid x_{t+1})} \tag{16}$$

$$= \frac{p(y \mid x_t)p_\theta(x_t \mid x_{t+1})}{p_\theta(y \mid x_{t+1})} \tag{17}$$

Distribution $p_\theta(y \mid x_{t+1})$ is independent of $x_t$, so we use the constant $N$ instead:

$$p_\theta(x_t \mid x_{t+1}, y) = \frac{1}{N}p_\theta(x_t \mid x_{t+1})p_\theta(y \mid x_t) \tag{18}$$

Taking the logarithm of both sides of the equation, it can be obtained that:

$$\log p_\theta(x_t|x_{t+1}, y) = \log(p_\theta(x_t|x_{t+1})p(y|x_t)) + N_1, \tag{19}$$

where $N_1 = -\log N$ can be seen as a normalizing constant. For the first term, the posterior $q(x_t|x_{t+1})$ used for sampling is hard to calculate directly. Therefore, we utilize the model with parameter $\theta$ pre-trained on five datasets to approximate the conditional probabilities.

Consider the expansion of $p_\theta(x_t|x_{t+1})$:

$$\log p_\theta(x_t|x_{t+1})p_\theta(y|x_t) = \log p_\theta(x_t|x_{t+1}) + \log p_\theta(y|x_t) \tag{20}$$

$$\approx -\frac{1}{2}(x_t - \mu_\theta)^T\Sigma_\theta^{-1}(x_t - \mu_\theta) + \log p_\theta(y|x_t) + C_1 \tag{21}$$

$C_1$ is a constant generated from the expansion of $p_\theta(x_t|x_{t+1})$, where

$$C_1 = -\log\left((2\pi)^{\frac{n}{2}}(|\Sigma_\theta|^{\frac{1}{2}})\right) = -\frac{n}{2}\log 2\pi - \frac{1}{2}\log|\Sigma_\theta| \tag{22}$$

Term $\log p_\theta(y|x_t)$ reflects the guidance from blurry map $y$ integrated in the reverse process of sampling $x_{t-1}$. The introduction of distance metric $\mathcal{L}$ and guidance scale $s$ is exploited to characterize the value of $\log p_\theta(y|x_t)$ as a heuristic algorithms:

$$\log p_\theta(y|x_t) = -s\mathcal{L}(\mathcal{K}_\varphi^t(\tilde{x}_0), y), \tag{23}$$

where $\tilde{x}_0$ is calculated by estimating and eliminating the noise contained in $x_t$:

$$\tilde{x}_0 = \frac{x_t}{\sqrt{\bar{\alpha}_t}} - \frac{\sqrt{1-\bar{\alpha}_t}\epsilon_\theta(x_t,t)}{\sqrt{\bar{\alpha}_t}} \tag{24}$$

Meanwhile, taylor expansion around $x_t = \mu_\theta$ can be leveraged to estimate $\log p_\theta(y|x_t)$. By taking the first two terms of the Taylor expansion, it can be estimated that:

$$\log p_\theta(y|x_t) \approx \log p_\theta(y|x_t)|_{x_t=\mu_\theta} + (x_t - \mu_\theta)^T\nabla_{x_t}\log p_\theta(y|x_t)|_{x_t=\mu_\theta} \tag{25}$$

$$= C_2 + (x_t - \mu_\theta)^T\nabla_{x_t}\log p_\theta(y|x_t)|_{x_t=\mu_\theta} \tag{26}$$

By combining the heuristic approximation formula and Taylor expansion mentioned above, we can derive the empirical formula for the guidance scale:

$$s = -\frac{(x_t - \mu)^T\nabla_{x_t}\log p_\theta(y|x_t)|_{x_t=\mu_\theta} + C_2}{\mathcal{L}(\mathcal{K}_\varphi^t(\tilde{x}_0), y)} \tag{27}$$

Among them, $\mu$ is obtained by pre-trained DDPM. $C_2$ is a minor constant that can be disregarded.

By incorporating Equation 26, we can further simplify Equation 21: (Taking $g$ to replace the gradient $\nabla_{x_t}\log p_\theta(y|x_t)|_{x_t=\mu_\theta}$)

$$\log p_\theta(x_t|x_{t+1})p_\theta(y|x_t) \approx -\frac{1}{2}(x_t - \mu_\theta)^T\Sigma_\theta^{-1}(x_t - \mu_\theta) + (x_t - \mu_\theta)^T g + C_1 + C_2 \tag{28}$$

$$= -\frac{1}{2}(x_t - \mu_\theta - \Sigma_\theta g)^T\Sigma_\theta^{-1}(x_t - \mu_\theta - \Sigma_\theta g) + \frac{1}{2}g^T\Sigma_\theta g + C_1 + C_2 \tag{29}$$

$$= \log p(z) + N_2, \quad z \sim \mathcal{N}(\mu_\theta + \Sigma_\theta g, \Sigma_\theta), \tag{30}$$

where $N_2 = \frac{1}{2}g^T\Sigma_\theta g + C_2$ is a constant related to $g$. Equation 30 is utilized to sample $x_{t-1}$ in every reverse process $t$.

### A.5 DETAILS OF EVALUATION THRESHOLDS

Specifically, the evaluated threshold $\tau$ for these datasets is

$$\tau = \begin{cases} 32.24 & kg/m^2 & \text{for SEVIR} \\ 30 & mm/h & \text{for HKO7, TAASRAD19, SRAD2018} \\ 40 & dbz & \text{for SCWDS CAP30 and SCWDS CR} \\ 47 & dbz & \text{for MeteoNet} \end{cases}$$

For the SEVIR dataset, we convert the VIL pixel into $R$ with the units of $kg/m^2$, which are the true units of VIL images, by the following rule (Veillette et al., 2020):

$$R(x) = \begin{cases} 0, & \text{if } x \leq 5 \\ \frac{x-2}{90.66}, & \text{if } 5 < x \leq 18 \\ \exp(\frac{x-83.9}{38.9}), & \text{if } 18 < x \leq 254 \end{cases}$$

Additionally, we transform the radar reflectivity values with dBZ unit in HKO7, TAASRAD19 , and SRAD2018 into rainfall intensity values ($mm/h$) using the Z-R relationship (Shi et al., 2017; Franch et al., 2020):

$$dBZ = 10 \log a + 10b \log R \tag{31}$$

where R is the rain-rate level, $a = 58.53$ and $b = 1.56$.

## A.6 DATASET DESCRIPTIONS

**SEVIR (Veillette et al., 2020)**    We use the data of NEXRAD which are processed into radar mosaic of Vertically Integrated Liquid (VIL) in SEVIR. It contains image sequences for over 10,000 weather events that cover 384 km x 384 km patches. These weather events are captured over the contiguous US (CONUS). The VIL image has a spatial resolution of 384 x 384. We use weather events in 2017 and 2018 for training and events in 2019 for validation and testing.

**HKO7 (Shi et al., 2017)**    In HKO7, the radar echoes are CAPPI (constant altitude plan position indicator) images covering a 512 km$^2$ region centered on Hong Kong, captured from an altitude of 2 km above sea level. The spatial resolution of data is 480x480. HKO7 contains observations from 2009 to 2015. Observations in the years 2009-2014 are split for training and validation, and data from 2015 are used for testing.

**TAASRAD19 (Shi et al., 2017)**    It collected radar echo images from 2010 to 2019 for a total of 1258 days using the maximum reflectivity of vertical, by the weather radar of the Trentino South Tyrol Region, in the Italian Alps. The images cover an area of 240 km with a spatial resolution of 480×480. The radar echoes from 2010-2018 are split into training part, and those of 2019 are used for validation and testing.

**Shanghai (Chen et al., 2020)**    It is a dataset that contains radar echoes represented by the CR (composite reflectivity) collected from the dual polarization Doppler radar located in Pudong, Shanghai. The raw data, spanning from October 2015 to July 2018, are interpolated to 0.01×0.01° longitude-latitude grids with a spatial resolution of 501×501. We use data from 2015 to 2017 for training, and 2018 for validation and testing.

**SRAD2018 (SRAD, 2018)**    This dataset covers Guangdong Province and Hong Kong during the flood season from 2010 to 2017. The size of radar images is 501 x 501 with a spatial resolution of 1 km. We follow  (SRAD, 2018) to split the dataset for training, validation, and testing.

**MeteoNet (Larvor & Berthomier, 2021)**    MeteoNet is a dataset covering two geographical zones, North-West and South-East of France, during three years, 2016 to 2018. The radar in MeteoNet has a spatial resolution of 0.01°. We crop the data to keep the top-left portion with a size of 400×400. We train the models with images from 2016 to 2017 and validate or test the models with images of 2018.

**SCWDS (Na et al., 2021)**    It is released by the National Meteorological Information Center of China, including a total of 9832 single-station strong precipitation processes across Jiangxi, Hubei, Anhui, Zhejiang, and Fujian from 2016 to 2018. We select 3km-CAPPI images as the modality of the SCWDS dataset. Each radar image in SCWDS covers 301$^2$ grids with a resolution of 0.01 x 0.01. Images from 2016 and 2017 are used for training, and those from 2018 are processed for validation and testing.

For each dataset, we randomly sample 500 sequences from the test split to evaluate the effectiveness of our proposed PostCast and other methods.

### A.7 IMPLEMENTING DETAILS OF ABLATION STUDY

The ablation study regarding the effectiveness of proposed zero-shot blur estimation mechanism and auto-scale gradient guidance strategy is conducted at the predictions of EarthFormer on HKO7 and SEVIR datasets. We utilize guidance scale $s = 3500$ for the fixed guidance scale model, as this setting closely aligns with the initial value of the guidance scale calculated by the auto-scale gradient guidance strategy for HKO7 and SEVIR datasets. Concurrently, the size of the blur kernel utilized in the ablation study is $9 \times 9$, and its initial value in the fixed-kernel model is randomly drawn from a Gaussian distribution with a mean of 0.6. This configuration is identical to that of the kernel's initial value in PostCast, thereby ensuring a fair comparison can be conducted in Table 4.

### A.8 RECOVERING THE BLURRY PREDICTIONS BY APPLYING BLUR KERNELS

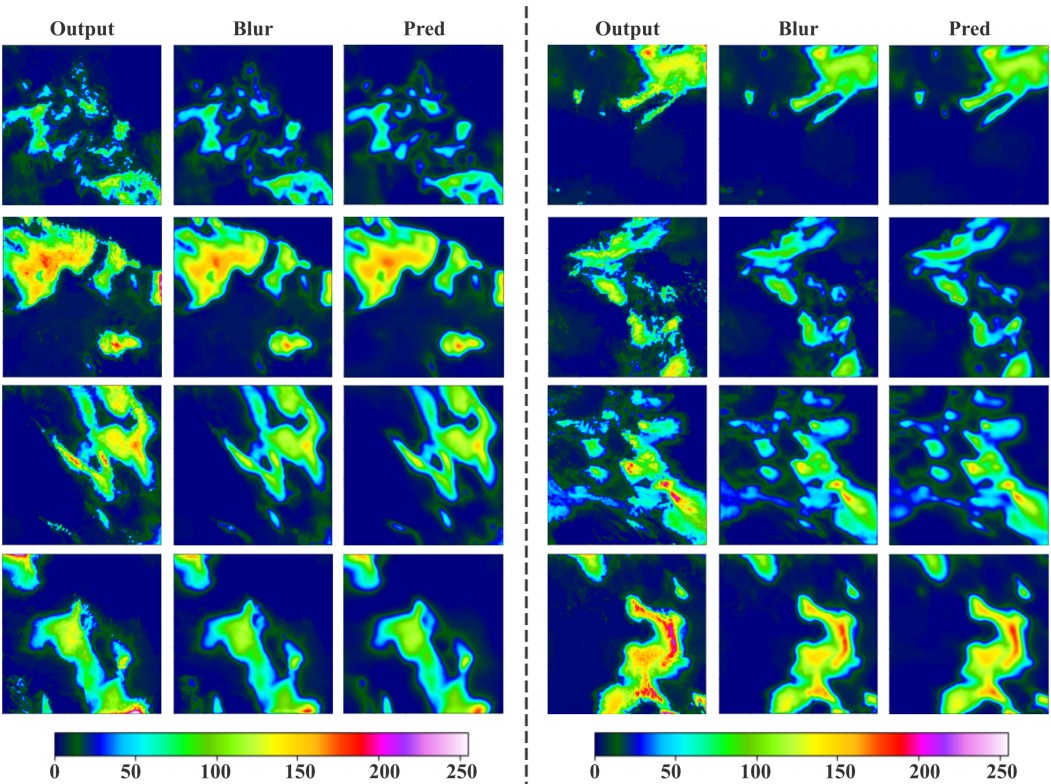

Figure 10: Visualization of a one-hour prediction by EarthFormer on the SEVIR dataset. From left to right, the three maps represent the PostCast output, output convolved with the optimizable blur kernel at time $t = 0$, and the prediction map. The convolved map closely resembles the prediction map, indicating that the model's blur kernel is effective in constructing and simulating the blur in the prediction map.

In this section, we validate whether the optimizable blur kernel effectively emulates the blur present in the prediction map by examining the correlation between the PostCast output and the derived blur kernel. The optimizable blur kernel is proposed to dynamically adjust its parameters in real-time during the reverse steps to simulate blur. As illustrated in Figure 10, the simulated blur map obtained by feeding the PostCast output after $T$ steps into the blur kernel exhibits no significant discrepancies with the actual prediction map. This indicates the efficacy of the blur kernel's parameters in faithfully replicating the blur inherent within the prediction map.

Table 5: The quantitive comparisons on average CSI scores, SSIM, and PSNR.

| Model | SEVIR | | | HKO7 | | | TAASRAD19 | | | Shanghai | | | SRAD2018 | | |
|---|---|---|---|---|---|---|---|---|---|---|---|---|---|---|---|
| | CSI-avg. | SSIM↑ | PSNR↑ | CSI-avg. | SSIM↑ | PSNR↑ | CSI-avg. | SSIM↑ | PSNR↑ | CSI-avg. | SSIM↑ | PSNR↑ | CSI-avg. | SSIM↑ | PSNR↑ |
| TAU | 0.294 | 0.666 | 21.98 | 0.271 | 0.642 | 21.59 | 0.179 | 0.799 | 27.38 | 0.221 | 0.739 | 21.52 | 0.198 | 0.635 | 27.51 |
| +ours | 0.305 | 0.648 | 21.92 | 0.266 | 0.682 | 20.66 | 0.192 | 0.708 | 25.27 | 0.231 | 0.750 | 20.60 | 0.231 | 0.750 | 20.60 |
| PredRNN | 0.288 | 0.673 | 22.30 | 0.241 | 0.666 | 20.99 | 0.151 | 0.785 | 26.95 | 0.182 | 0.772 | 21.37 | 0.183 | 0.800 | 27.23 |
| +ours | 0.299 | 0.659 | 21.44 | 0.238 | 0.684 | 19.83 | 0.177 | 0.709 | 25.11 | 0.205 | 0.772 | 19.79 | 0.205 | 0.772 | 19.79 |
| SimVP | 0.301 | 0.651 | 22.83 | 0.282 | 0.665 | 21.38 | 0.157 | 0.797 | 27.48 | 0.202 | 0.752 | 20.71 | 0.213 | 0.773 | 27.55 |
| +ours | 0.312 | 0.645 | 22.18 | 0.263 | 0.680 | 20.30 | 0.193 | 0.710 | 25.61 | 0.202 | 0.758 | 19.74 | 0.202 | 0.758 | 19.74 |
| EarthFormer | 0.288 | 0.673 | 22.79 | 0.270 | 0.617 | 21.53 | 0.169 | 0.793 | 26.59 | 0.205 | 0.772 | 21.09 | 0.201 | 0.699 | 27.23 |
| +ours | 0.308 | 0.653 | 21.73 | 0.270 | 0.656 | 20.65 | 0.182 | 0.713 | 25.21 | 0.217 | 0.769 | 18.57 | 0.217 | 0.769 | 18.57 |
| DiffCast | 0.270 | 0.628 | 20.27 | 0.263 | 0.594 | 19.90 | 0.167 | 0.759 | 25.15 | 0.205 | 0.674 | 19.94 | 0.198 | 0.627 | 21.65 |
| CasCast | 0.246 | 0.591 | 18.55 | 0.232 | 0.687 | 17.92 | 0.136 | 0.704 | 23.29 | 0.171 | 0.758 | 18.85 | 0.167 | 0.785 | 18.63 |
| DGMR | 0.110 | 0.169 | 15.34 | 0.133 | 0.088 | 14.71 | 0.058 | 0.267 | 14.66 | 0.066 | 0.074 | 12.55 | 0.068 | 0.066 | 10.58 |
| STRPM | 0.237 | 0.612 | 22.28 | 0.172 | 0.692 | 20.61 | 0.115 | 0.536 | 22.25 | 0.158 | 0.783 | 20.50 | 0.156 | 0.812 | 26.42 |
| DGP | 0.284 | 0.662 | 22.29 | 0.254 | 0.694 | 20.76 | 0.149 | 0.710 | 25.22 | 0.186 | 0.787 | 20.34 | 0.189 | 0.820 | 26.98 |

## A.9 MORE QUANTITATIVE RESULTS

More quantitative results are presented in Table 5. All the metrics are evaluated with in-domain settings which are the same as those in Section 4.3. After applying our PostCast, the CSI-avg. score increases in SEVIR, TAASRAD19, Shanghai, and SRAD2018, while slightly decrease in HKO7. It indicates that our PostCast boosts the accuracy of precipitation nowcasting in most cases. Image quality scores SSIM and PSNR are also computed. However, the PSNR and SSIM scores often penalize synthetic high-frequency details as demonstrated in Google SR3 (Saharia et al., 2022b), such as for the small-weather systems in weather prediction, the shape and position of local patterns may slightly deviate from the ground truth which may also result in inferior PSNR and SSIM scores but could be tolerated in applications. Moreover, these scores are not strongly related to nowcasting skills evaluated by the CSI. As a postprocessing method of precipitation nowcasting, our PostCast mainly focuses on improving meteorological metrics such as CSI and HSS.

## A.10 MORE VISUALIZATION RESULTS

We provide more visualizations in Figure 12 11 13 14 15 16 17.

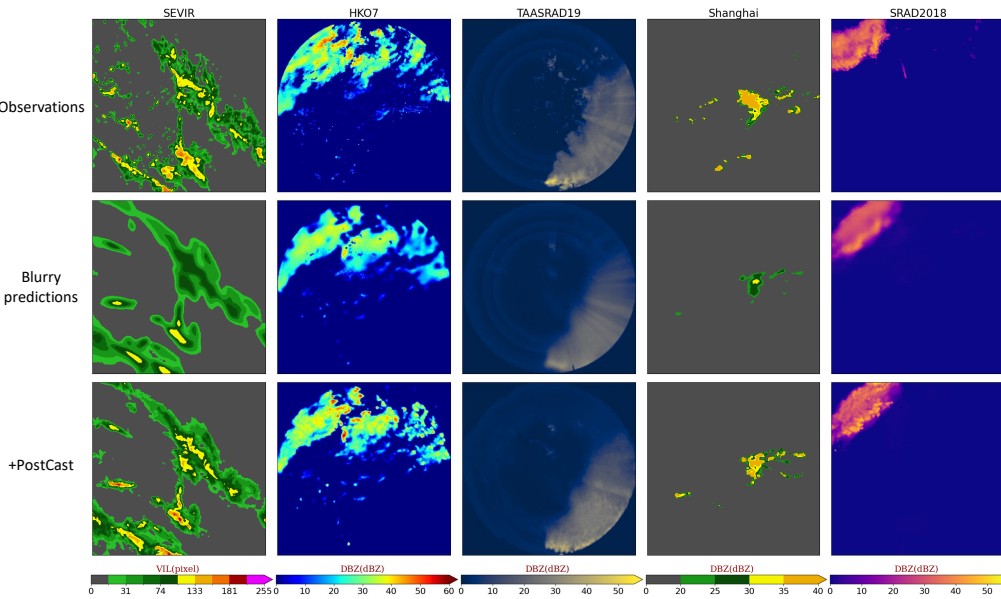

Figure 11: Visualization of applying our PostCast on 5 datasets at time step 12 when the spatiotemporal prediction model is PredRNN.

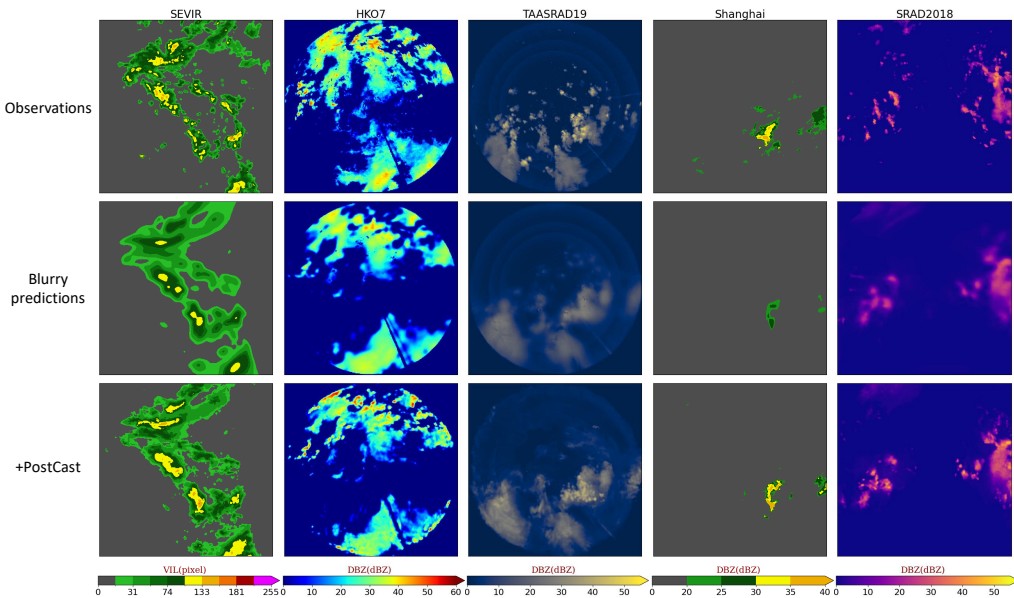

Figure 12: Visualization of applying our PostCast on 5 datasets at time step 12 when the spatiotemporal prediction model is SimVP.

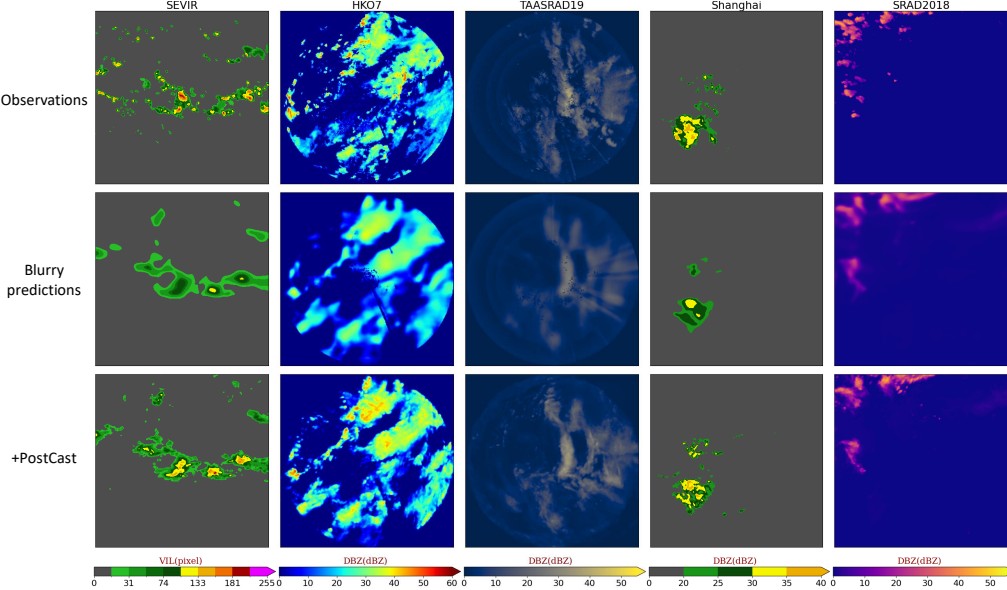

Figure 13: Visualization of applying our PostCast on 5 datasets at time step 12 when the spatiotemporal prediction model is EarthFormer.

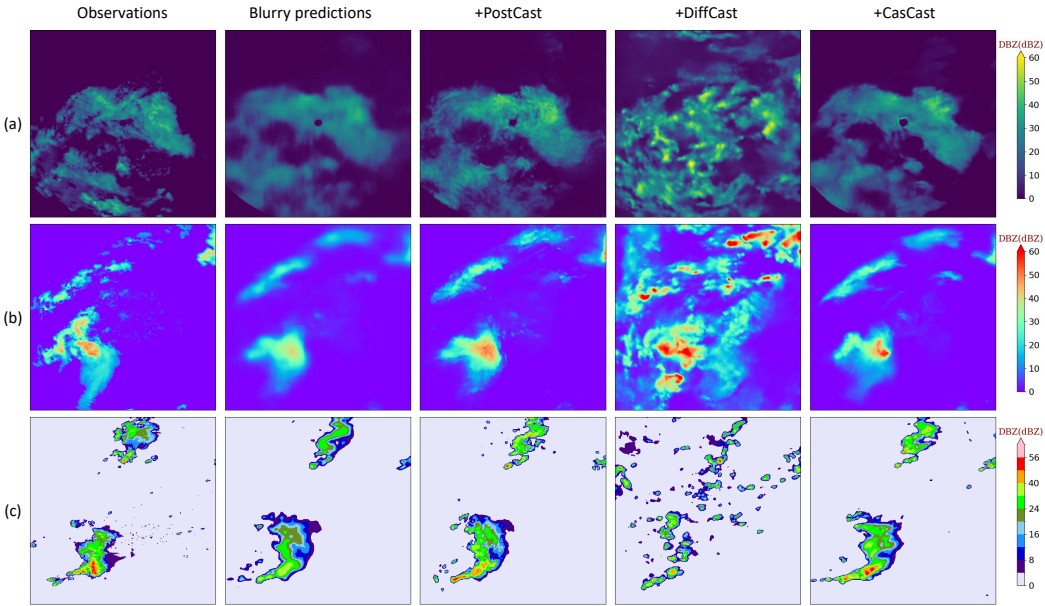

Figure 14: Visualization of applying our PostCast on the out-of-distribution datasets at time step 12 when the spatiotemporal prediction model is TAU. **(a)**: SCWDS CAP30. **(b)**: SCWDS CR. **(c)**: MeteoNet.

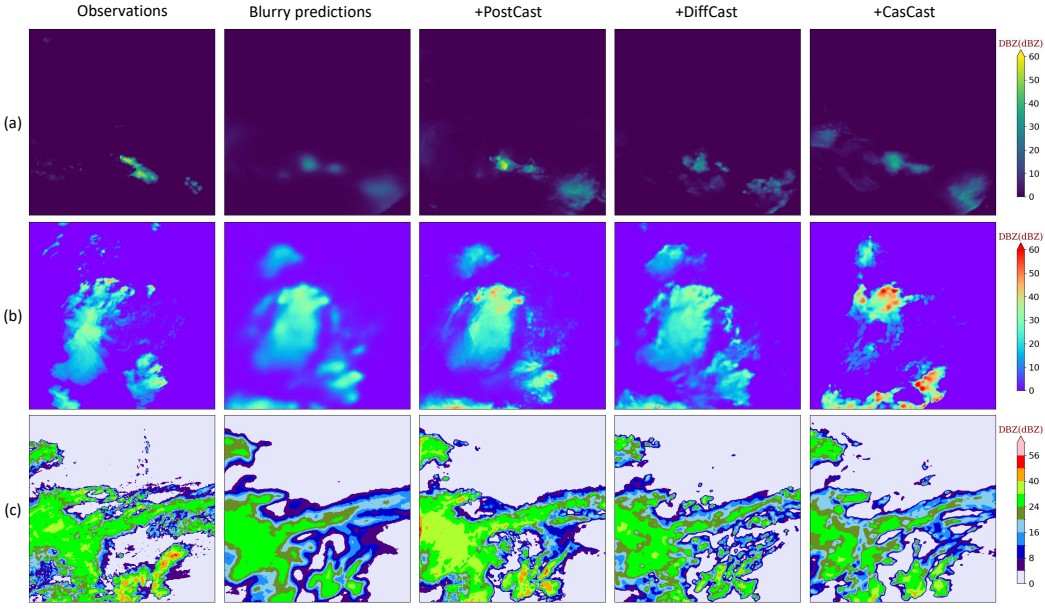

Figure 15: Visualization of applying our PostCast on the out-of-distribution datasets at time step 12 when the spatiotemporal prediction model is PredRNN. **(a)**: SCWDS CAP30. **(b)**: SCWDS CR. **(c)**: MeteoNet.

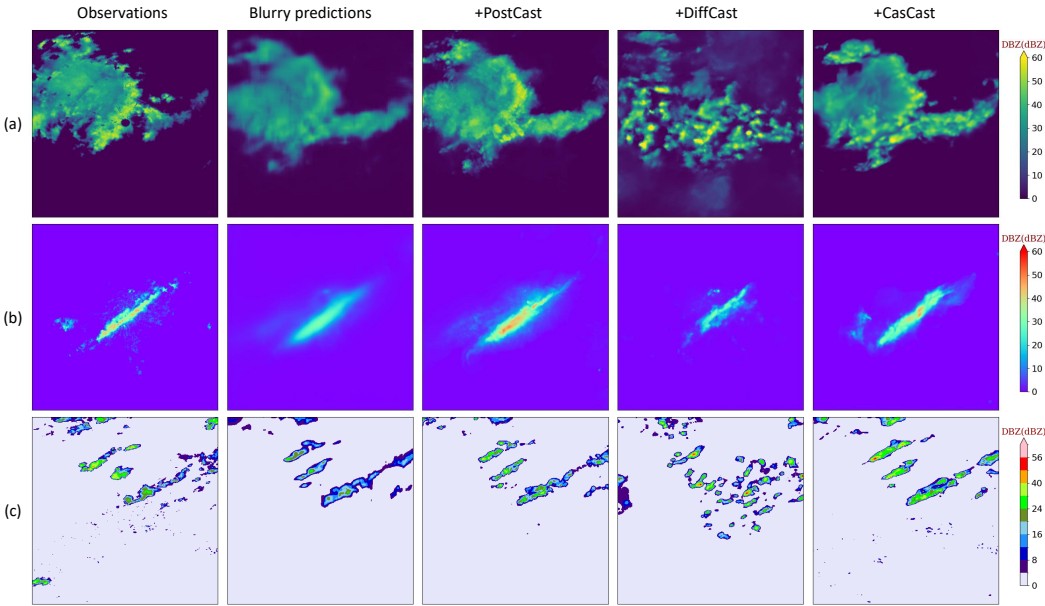

Figure 16: Visualization of applying our PostCast on the out-of-distribution datasets at time step 12 when the spatiotemporal prediction model is SimVP. **(a)**: SCWDS CAP30. **(b)**: SCWDS CR. **(c)**: MeteoNet.

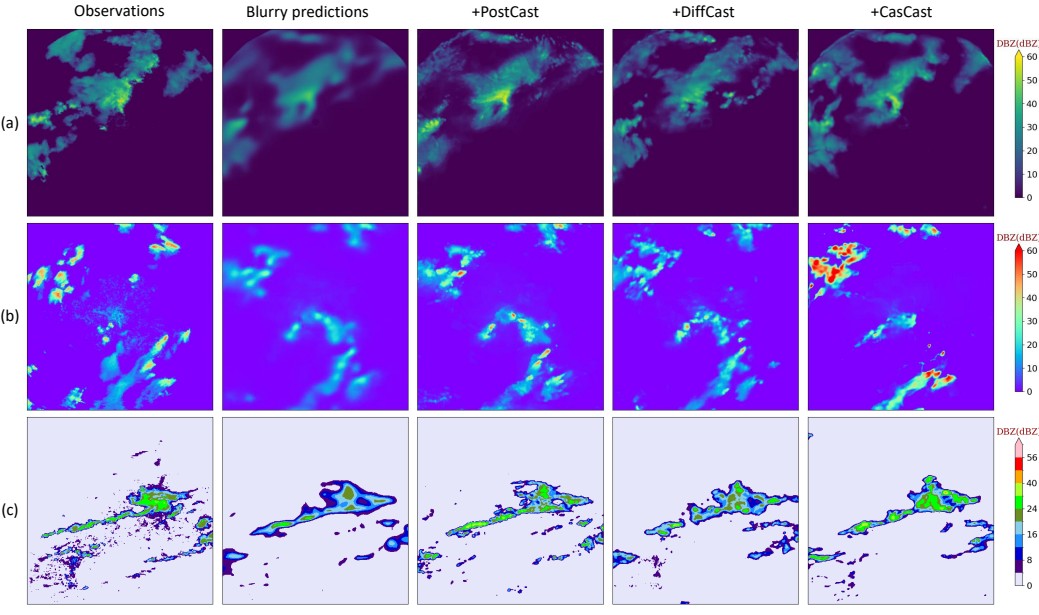

Figure 17: Visualization of applying our PostCast on the out-of-distribution datasets at time step 12 when the spatiotemporal prediction model is EarthFormer. **(a)**: SCWDS CAP30. **(b)**: SCWDS CR. **(c)**: MeteoNet..

