# OpenReview forum: "PostCast: Generalizable Postprocessing for Precipitation Nowcasting via Unsupervised Blurriness Modeling"
_ICLR.cc/2025/Conference — ICLR 2025 Poster_

### Official Review · Reviewer_CFdX · 2024-11-02

**Soundness:** 3
**Presentation:** 3
**Contribution:** 3
**Rating:** 8
**Confidence:** 3

**Summary:**

The paper proposes an unsupervised postprocessing method for precipitation nowcasting outputs to remove blurriness from these predictions, especially at longer lead times. This can be important to obtain accurate extreme precipitation events. The blurriness is eliminated with an unconditioned denoising diffusion probabilistic model guided by blurry predictions. The authors introduce a zero-shot kernel estimation method and an auto-scale denoise guidance strategy and show that their proposed framework generalizes to many blurriness modes across different datasets and varying lead times.

**Strengths:**

- The problem is well-motivated. An unsupervised method alleviates the need for labels and the model is no longer limited to the blur modes seen during training. Given that this framework is independent of the spatiotemporal prediction model and can be added on top of any prediction model, it makes this work very useful and widely applicable.
- The results strongly demonstrate the ability of their proposed method. The experiments are extensive and prove PostCast’s effectiveness in recovering weather patterns across multiple prediction models, datasets, and lead times.

**Weaknesses:**

- The "method" section could be more organized, creating subsections could help. Moreover, the “zero-shot kernel estimation mechanism” and “auto-scale denoise guidance strategy”, should be explained in more detail, as these are the main contributions this paper makes.
- The authors should justify using "CSI" (Critical Success Index) scores as the only evaluation metric. Adding another metric such as “Recall” (True positive rate) can complement the current evaluation and make it more comprehensive.

**Questions:**

The current metric/score mainly evaluates whether the small-scale features are recovered and extreme events are captured well after deblurring. However, isn’t it also important to assess how closely the nowcasting outputs, after deblurring with this methodology, match the ground truth, generally? If we don’t threshold the pixel values in this case and compute an error metric such as MSE and PSNR, that could also be informative. The authors can consider adding these scores.

---

> ### Author Response · Authors · 2024-11-29
> **Answer of W1 To Reviewer CFdX**
>
> Firstly, we want to sincerely thank the reviewer for the feedback and for acknowledging the strengths of this paper.
>
> We address each weakness and question below:
>
> >**W1:  The organization of "method" section.**
>
>
> **A1**: Thank you for your valuable feedback. In order to improve the organization of the "method" section, following your advice, we have created subsections for the “zero-shot kernel estimation mechanism” and “auto-scale denoise guidance strategy”. Moreover, to improve the clarity, we have also added more details into **Appendix A.4** and carefully created links in the "method" section. The revised sections are highlighted in blue.

---

> ### Author Response · Authors · 2024-11-29
> **Answer of W2 & Q1 To Reviewer CFdX**
>
> > **W2 & Q1: Additional metrics for a more comprehensive evaluation.**
>
>
> **A2**: Thanks for your valuable suggestion. As you suggested, we have added more evaluation metrics to achieve a more comprehensive evaluation, including meteorological metrics HSS, POD(Recall), CSI-avg. and image metrics PSNR and SSIM in the revised paper (see Table 1, Table 3, and Table 5 in the revised paper). These additional metrics further validate the effectiveness of our method. Part of the results are directly presented in the tables below:
>
> ### SEVIR
> | model       | CSI↑  | HSS↑  | POD↑  | SSIM↑ | PSNR↑ |
> |-------------|-------|-------|-------|-------|-------|
> | TAU         | 0.294 | 0.383 | 0.372 | 0.666 | 21.98 |
> | +ours       | 0.305 | 0.402 | 0.438 | 0.648 | 21.92 |
> | PredRNN     | 0.288 | 0.378 | 0.358 | 0.673 | 22.30 |
> | +ours       | 0.299 | 0.397 | 0.432 | 0.659 | 21.44 |
> | SimVP       | 0.301 | 0.389 | 0.385 | 0.651 | 22.83 |
> | +ours       | 0.312 | 0.409 | 0.462 | 0.645 | 22.18 |
> | EarthFormer | 0.288 | 0.374 | 0.357 | 0.673 | 22.79 |
> | +ours       | 0.308 | 0.403 | 0.427 | 0.653 | 21.73 |
> | DiffCast    | 0.270 | 0.362 | 0.378 | 0.628 | 20.27 |
> | CasCast     | 0.246 | 0.335 | 0.422 | 0.591 | 18.55 |
>
>
> ### HKO7
> | model       | CSI↑  | HSS↑  | POD↑  | SSIM↑ | PSNR↑ |
> |-------------|-------|-------|-------|-------|-------|
> | TAU         | 0.271 | 0.390 | 0.332 | 0.642 | 21.59 |
> | +ours       | 0.266 | 0.386 | 0.389 | 0.682 | 20.66 |
> | PredRNN     | 0.241 | 0.352 | 0.304 | 0.666 | 20.99 |
> | +ours       | 0.238 | 0.350 | 0.371 | 0.684 | 19.83 |
> | SimVP       | 0.282 | 0.409 | 0.358 | 0.665 | 21.38 |
> | +ours       | 0.263 | 0.385 | 0.409 | 0.680 | 20.30 |
> | EarthFormer | 0.270 | 0.390 | 0.334 | 0.617 | 21.53 |
> | +ours       | 0.270 | 0.392 | 0.395 | 0.656 | 20.65 |
> | DiffCast    | 0.263 | 0.385 | 0.375 | 0.594 | 19.90 |
> | CasCast     | 0.232 | 0.343 | 0.454 | 0.687 | 17.92 |
>
> ### TAASRAD19
>
> | model       | CSI↑  | HSS↑  | POD↑  | SSIM↑ | PSNR↑ |
> |-------------|-------|-------|-------|-------|-------|
> | TAU         | 0.179 | 0.276 | 0.220 | 0.799 | 27.38 |
> | +ours       | 0.192 | 0.300 | 0.311 | 0.708 | 25.27 |
> | PredRNN     | 0.151 | 0.237 | 0.178 | 0.785 | 26.95 |
> | +ours       | 0.177 | 0.279 | 0.275 | 0.709 | 25.11 |
> | SimVP       | 0.157 | 0.242 | 0.181 | 0.797 | 27.48 |
> | +ours       | 0.193 | 0.298 | 0.275 | 0.710 | 25.61 |
> | EarthFormer | 0.169 | 0.266 | 0.204 | 0.793 | 26.59 |
> | +ours       | 0.182 | 0.286 | 0.283 | 0.713 | 25.21 |
> | DiffCast    | 0.167 | 0.267 | 0.260 | 0.759 | 25.15 |
> | CasCast     | 0.136 | 0.221 | 0.301 | 0.704 | 23.29 |
>
> ### Shanghai
> | model       | CSI↑  | HSS↑  | POD↑  | SSIM↑ | PSNR↑ |
> |-------------|-------|-------|-------|-------|-------|
> | TAU         | 0.221 | 0.323 | 0.277 | 0.739 | 21.52 |
> | +ours       | 0.231 | 0.338 | 0.421 | 0.750 | 20.60 |
> | PredRNN     | 0.182 | 0.268 | 0.216 | 0.772 | 21.37 |
> | +ours       | 0.205 | 0.303 | 0.349 | 0.772 | 19.79 |
> | SimVP       | 0.202 | 0.303 | 0.258 | 0.752 | 20.71 |
> | +ours       | 0.202 | 0.303 | 0.390 | 0.758 | 19.74 |
> | EarthFormer | 0.205 | 0.304 | 0.253 | 0.772 | 21.09 |
> | +ours       | 0.217 | 0.321 | 0.396 | 0.769 | 18.57 |
> | DiffCast    | 0.205 | 0.309 | 0.282 | 0.674 | 19.94 |
> | CasCast     | 0.171 | 0.259 | 0.336 | 0.758 | 18.85 |
>
> ### SRAD2018
> | model       | CSI↑  | HSS↑  | POD↑  | SSIM↑ | PSNR↑ |
> |-------------|-------|-------|-------|-------|-------|
> | TAU         | 0.198 | 0.308 | 0.229 | 0.635 | 27.51 |
> | +ours       | 0.231 | 0.338 | 0.421 | 0.750 | 20.60 |
> | PredRNN     | 0.183 | 0.289 | 0.221 | 0.800 | 27.23 |
> | +ours       | 0.205 | 0.303 | 0.349 | 0.772 | 19.79 |
> | SimVP       | 0.213 | 0.331 | 0.258 | 0.773 | 27.55 |
> | +ours       | 0.202 | 0.303 | 0.390 | 0.758 | 19.74 |
> | EarthFormer | 0.201 | 0.311 | 0.244 | 0.699 | 27.23 |
> | +ours       | 0.217 | 0.321 | 0.396 | 0.769 | 18.57 |
> | DiffCast    | 0.198 | 0.313 | 0.307 | 0.627 | 21.65 |
> | CasCast     | 0.167 | 0.269 | 0.333 | 0.785 | 18.63 |

---

> ### Author Response · Authors · 2024-11-29
> **Answer of W2 & Q1 To Reviewer CFdX (continue)**
>
> ### SCWDS CAP30
> | model       | CSI↑  | HSS↑  | POD↑  | SSIM↑ | PSNR↑ |
> |-------------|-------|-------|-------|-------|-------|
> | TAU         | 0.220 | 0.312 | 0.280 | 0.589 | 20.60 |
> | +ours       | 0.245 | 0.345 | 0.428 | 0.613 | 19.48 |
> | PredRNN     | 0.170 | 0.239 | 0.203 | 0.573 | 20.17 |
> | +ours       | 0.234 | 0.331 | 0.351 | 0.592 | 19.25 |
> | SimVP       | 0.221 | 0.312 | 0.276 | 0.542 | 19.79 |
> | +ours       | 0.261 | 0.367 | 0.436 | 0.579 | 18.17 |
> | EarthFormer | 0.210 | 0.298 | 0.258 | 0.596 | 20.29 |
> | +ours       | 0.248 | 0.350 | 0.404 | 0.612 | 19.35 |
> | DiffCast    | 0.209 | 0.299 | 0.278 | 0.583 | 18.91 |
> | CasCast     | 0.205 | 0.350 | 0.404 | 0.645 | 18.24 |
>
> ### SCWDS CR
> | model       | CSI↑  | HSS↑  | POD↑  | SSIM↑ | PSNR↑ |
> |-------------|-------|-------|-------|-------|-------|
> | TAU         | 0.308 | 0.413 | 0.384 | 0.441 | 17.64 |
> | +ours       | 0.336 | 0.444 | 0.549 | 0.448 | 16.52 |
> | PredRNN     | 0.262 | 0.351 | 0.323 | 0.466 | 17.39 |
> | +ours       | 0.305 | 0.405 | 0.485 | 0.467 | 16.27 |
> | SimVP       | 0.309 | 0.410 | 0.373 | 0.388 | 17.94 |
> | +ours       | 0.349 | 0.459 | 0.545 | 0.420 | 17.55 |
> | EarthFormer | 0.314 | 0.417 | 0.406 | 0.346 | 17.67 |
> | +ours       | 0.337 | 0.444 | 0.570 | 0.385 | 16.74 |
> | DiffCast    | 0.304 | 0.407 | 0.417 | 0.387 | 16.48 |
> | CasCast     | 0.282 | 0.381 | 0.383 | 0.401 | 16.42 |
>
> ### MeteoNet
> | model       | CSI↑  | HSS↑  | POD↑  | SSIM↑ | PSNR↑ |
> |-------------|-------|-------|-------|-------|-------|
> | TAU         | 0.185 | 0.272 | 0.240 | 0.647 | 15.83 |
> | +ours       | 0.195 | 0.288 | 0.344 | 0.664 | 14.55 |
> | PredRNN     | 0.157 | 0.230 | 0.190 | 0.667 | 16.52 |
> | +ours       | 0.191 | 0.283 | 0.298 | 0.674 | 15.54 |
> | SimVP       | 0.195 | 0.281 | 0.245 | 0.672 | 16.15 |
> | +ours       | 0.218 | 0.316 | 0.391 | 0.665 | 15.05 |
> | EarthFormer | 0.175 | 0.259 | 0.219 | 0.660 | 16.00 |
> | +ours       | 0.194 | 0.287 | 0.320 | 0.674 | 15.03 |
> | DiffCast    | 0.176 | 0.263 | 0.243 | 0.592 | 14.94 |
> | CasCast     | 0.176 | 0.266 | 0.265 | 0.701 | 14.59 |
>
> Additionally, SSIM and PSNR often penalize high-frequency details as demonstrated in Google SR3 [Ref.1], such as for the small-weather systems in precipitation recovered by our PostCast. As a method for postprocessing for precipitation nowcasting, our PostCast mainly focuses on increasing the quality of predictions in a meteorological sense.
>
> [Ref.1] Chitwan Saharia, Jonathan Ho, William Chan, Tim Salimans, David J Fleet, and Mohammad Norouzi. Image super-resolution via iterative refinement. IEEE Transactions on Pattern Analysis and Machine Intelligence, 45(4):4713–4726, 2022b.

---

> ### Author Response · Authors · 2024-11-29
> **Answer of Q1 To Reviewer CFdX**
>
> > **Q1: The alignment between the nowcasting outputs after deblurring and the ground truth.**
>
>
> **A3**: Thanks for your valuable question. The degree of match between the nowcasting outputs, after deblurring with this methodology, and the ground truth can be assessed both qualitatively and quantitatively. Qualitatively, Figure 11-18 in **Appendix.A.10** shows that our PostCast provides the prediction with small weather patterns that better match the ground truth, and aligns the pixel value distribution of predictions closer to the ground truth. Quantitatively, we opt to evaluate our PostCase on meteological metrics such as CSI, HSS, POD rather than image metrics such as MSE or PSNR. The reason is that MSE and PSNR are limited to handling small-scale weather patterns with high-frequency details. For instance, the pixel-level offsets of the predictions in heavy precipitation significantly increase the MSE value, which implies MSE prefers smooth predictions. For PSNR, it often penalizes high-frequency details as demonstrated in Google SR3 [Ref.1], such as for the small-weather systems in precipitation. Instead, we choose other methods to evaluate the degree of matching. In methodology, the degree of matching is evaluated by the average of meteological metrics such as CSI, HSS, POD across different threshold intervals, which are added to the revised paper (see Table 1, Table 3, and Table 5 in the revised manuscript). These scores demonstrate that our PostCast aligns the predictions well with the ground truth.
>
> [Ref.1] Chitwan Saharia, Jonathan Ho, William Chan, Tim Salimans, David J Fleet, and Mohammad Norouzi. Image super-resolution via iterative refinement. IEEE Transactions on Pattern Analysis and Machine Intelligence, 45(4):4713–4726, 2022b.

---

> > ### Comment · Reviewer_CFdX · 2024-11-29
> > **Score updated**
> >
> > I thank the authors for adding additional evaluations and updating the manuscript with more details to improve clarity. I also saw other reviewers' comments and author responses. Taking everything into account, I have decided to increase my score.

---

> > > ### Author Response · Authors · 2024-11-30
> > > **Thank You**
> > >
> > > Dear reviewer CFdX,
> > >
> > > Thank you for reading our reply immediately and raising the score. Your appreciation for our additional evaluations and updating in the revised manuscript is inspiring. To further support our paper, we will carefully release our code. Thank you once again for your recognition of our work and the valuable time you have invested in this review.
> > >
> > > Best regards,
> > >
> > > The Authors

---

> > > > ### Author Response · Authors · 2024-12-03
> > > > **Official Comment by Authors**
> > > >
> > > > Thank you for your recognition of our work. We have provided General Comments and a Revision Summary, and we appreciate your constructive feedback that has helped make our paper better. We also welcome your participation in further discussions regarding the General Comments.

---

### Official Review · Reviewer_iPfR · 2024-11-02

**Soundness:** 2
**Presentation:** 2
**Contribution:** 2
**Rating:** 6
**Confidence:** 4

**Summary:**

This paper presents a novel postprocessing approach for precipitation nowcasting, named PostCast. The core innovation of PostCast lies in utilizing an unconditional diffusion model (DDPM) to remove blurriness from precipitation predictions without requiring paired data of real observations and blurry predictions. The method introduces a zero-shot blur kernel estimation mechanism and an auto-scale denoise guidance strategy, enabling the model to adapt to various datasets, prediction lead times, and blur modes to generate sharper predictions. The authors conducted experiments on multiple precipitation datasets and forecast models, demonstrating PostCast’s effectiveness in improving prediction accuracy for extreme precipitation events. The paper proposes a highly generalizable and adaptive postprocessing framework that shows strong potential for broad applications in precipitation nowcasting tasks.

**Strengths:**

1. **Originality**: The primary innovation of PostCast lies in employing an unconditional diffusion model (DDPM) to remove blurriness from precipitation predictions without relying on paired data. By introducing zero-shot blur kernel estimation and an auto-scale denoise guidance strategy, PostCast adapts to various datasets, prediction lead times, and blur modes across different models. This unsupervised deblurring approach appears to be the first of its kind in the domain of precipitation nowcasting, demonstrating creative thinking.

2. **Completeness of Experiments**: The paper provides a well-designed experimental setup, covering multiple datasets and forecast models to showcase the generalizability and robustness of PostCast. The experiments include diverse evaluation metrics and extensive visual results, effectively validating the method’s performance in extreme precipitation event prediction.

**Weaknesses:**

1. **Incomplete Evaluation Metrics**: The paper relies only on CSI, lacking POD, FAR, and similar precipitation metrics, as well as image quality metrics like PSNR and SSIM. Including these in the appendix would strengthen the evaluation’s comprehensiveness.

2. **Insufficient Related Work Discussion**: The discussion of GAN methods in precipitation nowcasting is limited, and the introduction to Transformer-based approaches is also lacking. Expanding these sections would better contextualize PostCast’s contributions.

3. **Lack of Comparison with GAN Methods**: There is no direct comparison with GAN-based approaches in precipitation nowcasting. Adding such comparisons would provide a more complete assessment of PostCast’s performance.

**Questions:**

1. **Evaluation Metrics**: Could you add results for POD, FAR, and image quality metrics like PSNR and SSIM in the appendix to provide a more comprehensive evaluation of PostCast?

2. **Related Work**: Could you expand the related work section to discuss GAN and Transformer methods in precipitation nowcasting to better contextualize the background and contributions of PostCast?

3. **Comparison with GAN Methods**: Could you include comparisons with GAN-based methods in precipitation nowcasting to further validate PostCast's effectiveness?

---

> ### Author Response · Authors · 2024-11-29
> **Answer of W1 & Q1 To Reviewer iPfR**
>
> The authors would like to express their heartfelt appreciation to the reviewer for the valuable and encouraging feedback.
>
> > **W1 & Q1:  Limited Evaluation Metrics.**
>
> **A1**: We thank you for the invaluable suggestion. In order to provide a more comprehensive evaluation of PostCast, we have added three meteorological metrics including CSI-avg., HSS-avg., POD-avg., as well as the two image metrics, SSIM and PSNR. We replace FAR-avg. with HSS-avg., as FAR only considers TP and FP, without accounting for FN which is quite important for precipitation nowcasting because the underreporting of extreme precipitation can lead to significant losses. Instead, HSS is a more comprehensive metric taking account into all the TP, FP, and FN. The formulations are:
> $FAR = \frac{FP}{TP+FP}$, $HSS = \frac{2\times(TP\times TN - FN \times FP)}{(TP + FN)\times (FN+TN)+(TP+FP)\times (FP+TN)}$.
> The POD-avg. and HSS-avg. are added to Table 1 and Table 3 in the revised paper with blue text. Meanwhile, CSI-avg., SSIM and PSNR scores are recorded in Table 5 in Appendix.9. We also collect a part of these metrics in the table below:
>
>
> ### SEVIR
> | model       | CSI↑  | HSS↑  | POD↑  | SSIM↑ | PSNR↑ |
> |-------------|-------|-------|-------|-------|-------|
> | TAU         | 0.294 | 0.383 | 0.372 | 0.666 | 21.98 |
> | +ours       | 0.305 | 0.402 | 0.438 | 0.648 | 21.92 |
> | PredRNN     | 0.288 | 0.378 | 0.358 | 0.673 | 22.30 |
> | +ours       | 0.299 | 0.397 | 0.432 | 0.659 | 21.44 |
> | SimVP       | 0.301 | 0.389 | 0.385 | 0.651 | 22.83 |
> | +ours       | 0.312 | 0.409 | 0.462 | 0.645 | 22.18 |
> | EarthFormer | 0.288 | 0.374 | 0.357 | 0.673 | 22.79 |
> | +ours       | 0.308 | 0.403 | 0.427 | 0.653 | 21.73 |
> | DiffCast    | 0.270 | 0.362 | 0.378 | 0.628 | 20.27 |
> | CasCast     | 0.246 | 0.335 | 0.422 | 0.591 | 18.55 |
>
>
> ### HKO7
> | model       | CSI↑  | HSS↑  | POD↑  | SSIM↑ | PSNR↑ |
> |-------------|-------|-------|-------|-------|-------|
> | TAU         | 0.271 | 0.390 | 0.332 | 0.642 | 21.59 |
> | +ours       | 0.266 | 0.386 | 0.389 | 0.682 | 20.66 |
> | PredRNN     | 0.241 | 0.352 | 0.304 | 0.666 | 20.99 |
> | +ours       | 0.238 | 0.350 | 0.371 | 0.684 | 19.83 |
> | SimVP       | 0.282 | 0.409 | 0.358 | 0.665 | 21.38 |
> | +ours       | 0.263 | 0.385 | 0.409 | 0.680 | 20.30 |
> | EarthFormer | 0.270 | 0.390 | 0.334 | 0.617 | 21.53 |
> | +ours       | 0.270 | 0.392 | 0.395 | 0.656 | 20.65 |
> | DiffCast    | 0.263 | 0.385 | 0.375 | 0.594 | 19.90 |
> | CasCast     | 0.232 | 0.343 | 0.454 | 0.687 | 17.92 |
>
> ### TAASRAD19
>
> | model       | CSI↑  | HSS↑  | POD↑  | SSIM↑ | PSNR↑ |
> |-------------|-------|-------|-------|-------|-------|
> | TAU         | 0.179 | 0.276 | 0.220 | 0.799 | 27.38 |
> | +ours       | 0.192 | 0.300 | 0.311 | 0.708 | 25.27 |
> | PredRNN     | 0.151 | 0.237 | 0.178 | 0.785 | 26.95 |
> | +ours       | 0.177 | 0.279 | 0.275 | 0.709 | 25.11 |
> | SimVP       | 0.157 | 0.242 | 0.181 | 0.797 | 27.48 |
> | +ours       | 0.193 | 0.298 | 0.275 | 0.710 | 25.61 |
> | EarthFormer | 0.169 | 0.266 | 0.204 | 0.793 | 26.59 |
> | +ours       | 0.182 | 0.286 | 0.283 | 0.713 | 25.21 |
> | DiffCast    | 0.167 | 0.267 | 0.260 | 0.759 | 25.15 |
> | CasCast     | 0.136 | 0.221 | 0.301 | 0.704 | 23.29 |
>
>
> ### Shanghai
> | model       | CSI↑  | HSS↑  | POD↑  | SSIM↑ | PSNR↑ |
> |-------------|-------|-------|-------|-------|-------|
> | TAU         | 0.221 | 0.323 | 0.277 | 0.739 | 21.52 |
> | +ours       | 0.231 | 0.338 | 0.421 | 0.750 | 20.60 |
> | PredRNN     | 0.182 | 0.268 | 0.216 | 0.772 | 21.37 |
> | +ours       | 0.205 | 0.303 | 0.349 | 0.772 | 19.79 |
> | SimVP       | 0.202 | 0.303 | 0.258 | 0.752 | 20.71 |
> | +ours       | 0.202 | 0.303 | 0.390 | 0.758 | 19.74 |
> | EarthFormer | 0.205 | 0.304 | 0.253 | 0.772 | 21.09 |
> | +ours       | 0.217 | 0.321 | 0.396 | 0.769 | 18.57 |
> | DiffCast    | 0.205 | 0.309 | 0.282 | 0.674 | 19.94 |
> | CasCast     | 0.171 | 0.259 | 0.336 | 0.758 | 18.85 |
>
> ### SRAD2018
> | model       | CSI↑  | HSS↑  | POD↑  | SSIM↑ | PSNR↑ |
> |-------------|-------|-------|-------|-------|-------|
> | TAU         | 0.198 | 0.308 | 0.229 | 0.635 | 27.51 |
> | +ours       | 0.231 | 0.338 | 0.421 | 0.750 | 20.60 |
> | PredRNN     | 0.183 | 0.289 | 0.221 | 0.800 | 27.23 |
> | +ours       | 0.205 | 0.303 | 0.349 | 0.772 | 19.79 |
> | SimVP       | 0.213 | 0.331 | 0.258 | 0.773 | 27.55 |
> | +ours       | 0.202 | 0.303 | 0.390 | 0.758 | 19.74 |
> | EarthFormer | 0.201 | 0.311 | 0.244 | 0.699 | 27.23 |
> | +ours       | 0.217 | 0.321 | 0.396 | 0.769 | 18.57 |
> | DiffCast    | 0.198 | 0.313 | 0.307 | 0.627 | 21.65 |
> | CasCast     | 0.167 | 0.269 | 0.333 | 0.785 | 18.63 |

---

> ### Author Response · Authors · 2024-11-29
> **Answer of W1 & Q1 To Reviewer iPfR (continue)**
>
> ### SCWDS CAP30
> | model       | CSI↑  | HSS↑  | POD↑  | SSIM↑ | PSNR↑ |
> |-------------|-------|-------|-------|-------|-------|
> | TAU         | 0.220 | 0.312 | 0.280 | 0.589 | 20.60 |
> | +ours       | 0.245 | 0.345 | 0.428 | 0.613 | 19.48 |
> | PredRNN     | 0.170 | 0.239 | 0.203 | 0.573 | 20.17 |
> | +ours       | 0.234 | 0.331 | 0.351 | 0.592 | 19.25 |
> | SimVP       | 0.221 | 0.312 | 0.276 | 0.542 | 19.79 |
> | +ours       | 0.261 | 0.367 | 0.436 | 0.579 | 18.17 |
> | EarthFormer | 0.210 | 0.298 | 0.258 | 0.596 | 20.29 |
> | +ours       | 0.248 | 0.350 | 0.404 | 0.612 | 19.35 |
> | DiffCast    | 0.209 | 0.299 | 0.278 | 0.583 | 18.91 |
> | CasCast     | 0.205 | 0.350 | 0.404 | 0.645 | 18.24 |
>
> ### SCWDS CR
> | model       | CSI↑  | HSS↑  | POD↑  | SSIM↑ | PSNR↑ |
> |-------------|-------|-------|-------|-------|-------|
> | TAU         | 0.308 | 0.413 | 0.384 | 0.441 | 17.64 |
> | +ours       | 0.336 | 0.444 | 0.549 | 0.448 | 16.52 |
> | PredRNN     | 0.262 | 0.351 | 0.323 | 0.466 | 17.39 |
> | +ours       | 0.305 | 0.405 | 0.485 | 0.467 | 16.27 |
> | SimVP       | 0.309 | 0.410 | 0.373 | 0.388 | 17.94 |
> | +ours       | 0.349 | 0.459 | 0.545 | 0.420 | 17.55 |
> | EarthFormer | 0.314 | 0.417 | 0.406 | 0.346 | 17.67 |
> | +ours       | 0.337 | 0.444 | 0.570 | 0.385 | 16.74 |
> | DiffCast    | 0.304 | 0.407 | 0.417 | 0.387 | 16.48 |
> | CasCast     | 0.282 | 0.381 | 0.383 | 0.401 | 16.42 |
>
> ### MeteoNet
> | model       | CSI↑  | HSS↑  | POD↑  | SSIM↑ | PSNR↑ |
> |-------------|-------|-------|-------|-------|-------|
> | TAU         | 0.185 | 0.272 | 0.240 | 0.647 | 15.83 |
> | +ours       | 0.195 | 0.288 | 0.344 | 0.664 | 14.55 |
> | PredRNN     | 0.157 | 0.230 | 0.190 | 0.667 | 16.52 |
> | +ours       | 0.191 | 0.283 | 0.298 | 0.674 | 15.54 |
> | SimVP       | 0.195 | 0.281 | 0.245 | 0.672 | 16.15 |
> | +ours       | 0.218 | 0.316 | 0.391 | 0.665 | 15.05 |
> | EarthFormer | 0.175 | 0.259 | 0.219 | 0.660 | 16.00 |
> | +ours       | 0.194 | 0.287 | 0.320 | 0.674 | 15.03 |
> | DiffCast    | 0.176 | 0.263 | 0.243 | 0.592 | 14.94 |
> | CasCast     | 0.176 | 0.266 | 0.265 | 0.701 | 14.59 |
>
> On meteorological metrics CSI-avg., HSS-avg.,  and FAR-avg., our PostCast significantly surpasses DiffCast and CasCast. After applying our PostCast, these metrics are improved notably on most datasets. On image quality scores, SSIM and PSNR, our PostCast improved SSIM in some settings, such as HKO7 and TAU-SRAD2018, while the PSNR scores decreased. Such a phenomenon also appears when applying DiffCast and CasCast. It can be attributed to that the PSNR and SSIM scores often penalize synthetic high-frequency details as demonstrated in Google SR3 [Ref.1], such as for the small-weather systems in weather prediction, the shape and position of local patterns may slightly deviate from the ground truth which may also result in inferior PSNR and SSIM scores but could be tolerated in applications. Moreover, these scores are not strongly related to nowcasting skills evaluated by the meteorological metrics  CSI-avg. , HSS-avg. ,  and FAR-avg. As a postprocessing method of precipitation nowcasting, our PostCast mainly focuses on improving meteorological metrics
>
> [Ref.1 ] Chitwan Saharia, Jonathan Ho, William Chan, Tim Salimans, David J Fleet, and Mohammad Norouzi. Image super-resolution via iterative refinement. IEEE Transactions on Pattern Analysis and Machine Intelligence, 45(4):4713–4726, 2022b.

---

> ### Author Response · Authors · 2024-11-29
> **Answer of W2 & Q2 To Reviewer iPfR**
>
> >**W2 & Q2:  Enrich the Related Work.**
>
> **A2**: Thanks for your valuable suggestions. Indeed, discussing GAN-based and Transformer-based precipitation nowcasting methods could definitely better contextualize the background and contributions of PostCast. Therefore, we have discussed EarthFormer and DGMR  in the related work, which could be directly found as follows:
>
> Notable progress has been achieved by applying deep learning in precipitation nowcasting [Refs.1-5]. The initial attempts are deterministic methods focusing on spatiotemporal modeling. Researchers explore different spatiotemporal modeling structures such as RNN [Refs.6,7], CNN [Refs.8,9], and Transformer [Refs.10]. **Specifically, EarthFormer [Refs.10] is proposed to capture spatial-temporal features in earth system evolution by cuboid attention.** However, these methods have a shortage of blurry predictions, which hamper the nowcasting of extreme events. Probabilistic methods are proposed to alleviate blurriness [Refs.11,12,13]. **DGMR [Refs.12] applies GAN to produce realistic and spatio-temporally consistent predictions to reduce blurriness.** To further enhance precipitation nowcasting with accurate global movements, later methods combine blurry predictions with probabilistic models. DiffCast [Refs.14]), and CasCast [Refs.15] exploit how to generate small-scale weather pattern conditioning on blurry predictions. Although these deterministic-probabilistic coupling methods achieve both global accuracy and local details, they suffer from repeating training for different contexts and require blurry predictions as data for training. From the perspective of deblur, Our method is proposed to simplify the complex training process and enhance the generality for wide use in different contexts.
>
> [Ref.1] Kang Chen, et al. Fengwu: Pushing the skillful global medium-range weather forecast beyond 10 days lead. arXiv preprint arXiv:2304.02948, 2023.
>
> [Ref.2] Tao Han, et al. Fengwu-ghr: Learning the kilometer-scale medium-range global weather forecasting. arXiv preprint arXiv:2402.00059, 2024b.
>
> [Ref.3] Wanghan Xu, et al. Extremecast: Boosting extreme value prediction for global weather forecast. arXiv preprint arXiv:2402.01295, 2024.
>
> [Ref.4] Tao Han, et al. Weather-5k: A large-scale global station weather dataset towards comprehensive time-series forecasting benchmark. arXiv preprint arXiv:2406.14399, 2024a.
>
> [Ref.5] Junchao Gong, et al. Weatherformer: Empowering global numerical weather forecasting with space-time transformer, 2024. arXiv preprint arXiv:2409.16321, 2024.
>
> [Ref.6] Xingjian Shi, et al. Convolutional lstm network: A machine learning approach for precipitation nowcasting. Advances in Neural Information Processing Systems, 28, 2015.
>
> [Ref.7] Yunbo Wang, et al. Predrnn: A recurrent neural network for spatiotemporal predictive learning. IEEE Transactions on Pattern Analysis and Machine Intelligence, 45(2):2208–2225, 2022.
>
> [Ref.8] Zhangyang Gao, et al. In Proceedings of the IEEE/CVF Conference on Computer Vision and Pattern Recognition, pp. 3170–3180, 2022a.
>
> [Ref.9] Cheng Tan, et al. Temporal attention unit: Towards efficient spatiotemporal predictive learning. In Proceedings of the IEEE/CVF Conference on Computer Vision and Pattern Recognition, pp. 18770–18782, 2023.
>
> **[Ref.10] Zhihan Gao, et al. Earthformer: Exploring space-time transformers for earth system forecasting. Advances in Neural Information Processing Systems, 35:25390–25403, 2022b.**
>
> **[Ref.11] Suman Ravuri, et al. Skilful precipitation nowcasting using deep generative models of radar. Nature, 597(7878):672–677, 2021.**
>
> [Ref.12] Zhihan Gao, et al. Prediff: Precipitation nowcasting with latent diffusion models. Advances in Neural Information Processing Systems, 36, 2024.
>
> [Ref.13] Zewei Zhao, et al. Advancing realistic precipitation nowcasting with a spatiotemporal transformer-based denoising diffusion model. IEEE Transactions on Geoscience and Remote Sensing, 2024.
>
> [Ref.14] Demin Yu, et al. Diffcast: A unified framework via residual diffusion for precipitation nowcasting. In Proceedings of the IEEE/CVF Conference on Computer Vision and Pattern Recognition, pp. 27758–27767, 2024.
>
> [Ref.15] Junchao Gong, et al. Cascast: Skillful high-resolution precipitation nowcasting via cascaded modelling. In Forty-first International Conference on Machine Learning.

---

> ### Author Response · Authors · 2024-11-29
> **Answer of W3&Q3 To Reviewer iPfR**
>
> > **W3 & Q3:  Comparison with GAN Methods:**
>
> **A3**: Many thanks for your valuable suggestion. To further validate PostCast's effectiveness, we include comparisons with GAN-based methods in precipitation nowcasting including DGMR[Ref.1] and STRPM[Ref.2]. The results are added to Table 1 and Table 3 in our revised paper. We also select the comparisons of our PostCast with diffusion-based methods and GAN-based methods in the table below:
>
> ### SEVIR
> | model    | P1    | P4    | P16   | CSI↑  | HSS↑  | POD↑  |
> |----------|-------|-------|-------|-------|-------|-------|
> | ours     | 0.045 | 0.070 | 0.131 | 0.308 | 0.403 | 0.427 |
> | DiffCast | 0.049 | 0.070 | 0.186 | 0.270 | 0.362 | 0.378 |
> | CasCast  | 0.039 | 0.067 | 0.156 | 0.246 | 0.335 | 0.422 |
> | DGMR     | 0.003 | 0.010 | 0.062 | 0.110 | 0.122 | 0.235 |
> | STRPM    | 0.007 | 0.023 | 0.060 | 0.237 | 0.307 | 0.296 |
>
> ### HKO7
> | model    | P1    | P4    | P16   | CSI↑  | HSS↑  | POD↑  |
> |----------|-------|-------|-------|-------|-------|-------|
> | ours     | 0.066 | 0.125 | 0.257 | 0.270 | 0.392 | 0.395 |
> | DiffCast | 0.061 | 0.113 | 0.255 | 0.263 | 0.385 | 0.375 |
> | CasCast  | 0.054 | 0.108 | 0.235 | 0.232 | 0.343 | 0.454 |
> | DGMR     | 0.018 | 0.055 | 0.210 | 0.133 | 0.210 | 0.182 |
> | STRPM    | 0.010 | 0.027 | 0.078 | 0.172 | 0.263 | 0.196 |
>
> ### TAASRAD19
> | model    | P1    | P4    | P16   | CSI↑  | HSS↑  | POD↑  |
> |----------|-------|-------|-------|-------|-------|-------|
> | ours     | 0.044 | 0.067 | 0.143 | 0.713 | 0.286 | 0.283 |
> | DiffCast | 0.044 | 0.076 | 0.174 | 0.759 | 0.267 | 0.260 |
> | CasCast  | 0.040 | 0.064 | 0.128 | 0.704 | 0.221 | 0.301 |
> | DGMR     | 0.015 | 0.038 | 0.120 | 0.267 | 0.097 | 0.091 |
> | STRPM    | 0.005 | 0.016 | 0.054 | 0.536 | 0.186 | 0.138 |
>
> ### Shanghai
> | model    | P1    | P4    | P16   | CSI↑  | HSS↑  | POD↑  |
> |----------|-------|-------|-------|-------|-------|-------|
> | ours     | 0.048 | 0.098 | 0.226 | 0.769 | 0.321 | 0.396 |
> | DiffCast | 0.050 | 0.097 | 0.218 | 0.674 | 0.309 | 0.282 |
> | CasCast  | 0.034 | 0.068 | 0.167 | 0.758 | 0.259 | 0.336 |
> | DGMR     | 0.007 | 0.028 | 0.132 | 0.074 | 0.105 | 0.103 |
> | STRPM    | 0.121 | 0.428 | 0.128 | 0.783 | 0.236 | 0.201 |
>
> ### SRAD2018
> | model    | P1    | P4    | P16   | CSI↑  | HSS↑  | POD↑  |
> |----------|-------|-------|-------|-------|-------|-------|
> | ours     | 0.095 | 0.155 | 0.276 | 0.769 | 0.321 | 0.396 |
> | DiffCast | 0.071 | 0.124 | 0.257 | 0.627 | 0.313 | 0.307 |
> | CasCast  | 0.061 | 0.109 | 0.240 | 0.785 | 0.269 | 0.333 |
> | DGMR     | 0.022 | 0.066 | 0.213 | 0.066 | 0.114 | 0.147 |
> | STRPM    | 0.034 | 0.076 | 0.171 | 0.812 | 0.251 | 0.197 |
>
> ### SCWDS CAP30
> | model    | P1    | P4    | P16   | CSI↑  | HSS↑  | POD↑  |
> |----------|-------|-------|-------|-------|-------|-------|
> | ours     | 0.070 | 0.117 | 0.241 | 0.249 | 0.350 | 0.404 |
> | DiffCast | 0.041 | 0.071 | 0.175 | 0.210 | 0.299 | 0.278 |
> | CasCast  | 0.050 | 0.089 | 0.190 | 0.206 | 0.296 | 0.287 |
> | DGMR     | 0.018 | 0.048 | 0.160 | 0.111 | 0.161 | 0.153 |
> | STRPM    | 0.014 | 0.049 | 0.160 | 0.163 | 0.234 | 0.197 |
>
> ### SCWDS CR
> | model    | P1    | P4    | P16   | CSI↑  | HSS↑  | POD↑  |
> |----------|-------|-------|-------|-------|-------|-------|
> | ours     | 0.141 | 0.211 | 0.326 | 0.338 | 0.444 | 0.570 |
> | DiffCast | 0.101 | 0.144 | 0.268 | 0.305 | 0.407 | 0.417 |
> | CasCast  | 0.100 | 0.130 | 0.223 | 0.283 | 0.381 | 0.383 |
> | DGMR     | 0.039 | 0.090 | 0.240 | 0.154 | 0.207 | 0.208 |
> | STRPM    | 0.029 | 0.080 | 0.201 | 0.220 | 0.296 | 0.264 |
>
> ### MeteoNet
> | model    | P1    | P4    | P16   | CSI↑  | HSS↑  | POD↑  |
> |----------|-------|-------|-------|-------|-------|-------|
> | ours     | 0.019 | 0.058 | 0.164 | 0.194 | 0.287 | 0.320 |
> | DiffCast | 0.009 | 0.029 | 0.096 | 0.176 | 0.263 | 0.243 |
> | CasCast  | 0.019 | 0.055 | 0.159 | 0.177 | 0.266 | 0.265 |
> | DGMR     | 0.019 | 0.057 | 0.192 | 0.078 | 0.123 | 0.131 |
> | STRPM    | 0.014 | 0.046 | 0.145 | 0.127 | 0.192 | 0.155 |
>
> As shown in these tables, our PostCast outperforms all GAN-based models in both in-domain and out-of-domain settings on extreme precipitation nowcasting evaluated by P1, P4, P16, and precipitation with different intensities by CSI, HSS and POD scores. These comprehensive evaluations could further validate the effectiveness of our PostCast.
>
> [Ref.1] Ravuri, Suman, et al. "Skilful precipitation nowcasting using deep generative models of radar." Nature 597.7878 (2021): 672-677.
>
> [Ref.2] Chang, Zheng, et al. "Strpm: A spatiotemporal residual predictive model for high-resolution video prediction." Proceedings of the IEEE/CVF conference on computer vision and pattern recognition. 2022.

---

> ### Comment · Reviewer_iPfR · 2024-11-30
>
> Thank you to the authors for addressing my concerns regarding W1 and W3. I truly appreciate the detailed experiments provided. I have adjusted my scores accordingly. However, the issue with W2 has not been adequately resolved. If the authors could address this, I would be willing to further increase my score.
>
> I would like to clarify that my main concern of W2 is that the related work section's discussion on **PRECIPITATION NOWCASTING** remains overly brief.
>
> For example, the discussion on **Transformer-based methods** is limited to EarthFormer, with no further expansion. Key works such as TCTN [Ref.20] and ConvTransformer [Ref.21], which are closely related to ConvLSTM, should be cited to provide a more comprehensive overview.
>
> In contrast, the discussion on **GAN-based methods** requires more depth, including an analysis of their strengths and limitations. Works like Two-stage UA-GAN [Ref.16], CLGAN [Refs.17,19], and Luo et al. [Ref.18] should be discussed in greater detail to highlight their relevance and provide critical insights into how they address precipitation nowcasting challenges.
>
> [Ref.16] Liujia Xu, Dan Niu, Tianbao Zhang, Pengju Chen, Xunlai Chen, Yinghao Li. Two-stage UA-GAN for precipitation nowcasting. Remote Sensing, 14(23):5948, 2022.
>
> [Ref.17] Yan Ji, Bing Gong, Michael Langguth, Amirpasha Mozaffari, Xiefei Zhi. CLGAN: a generative adversarial network (GAN)-based video prediction model for precipitation nowcasting. Geoscientific Model Development, 16(10):2737–2752, 2023.
>
> [Ref.18] Chuyao Luo, Xutao Li, Yunming Ye, Shanshan Feng, Michael K. Ng. Experimental study on generative adversarial network for precipitation nowcasting. IEEE Transactions on Geoscience and Remote Sensing, 60:1–20, 2022.
>
> [Ref.19] Yan Ji, Bing Gong, Michael Langguth, Amirpasha Mozaffari, Xiefei Zhi. CLGAN: A GAN-based video prediction model for precipitation nowcasting. EGUsphere, 2022:1–23, 2022.
>
> [Ref.20] Ziao Yang, Xiangrui Yang, Qifeng Lin. TCTN: A 3D-temporal convolutional transformer network for spatiotemporal predictive learning. arXiv preprint arXiv:2112.01085, 2021.
>
> [Ref.21] Zhouyong Liu, Shun Luo, Wubin Li, Jingben Lu, Yufan Wu, Shilei Sun, Chunguo Li, Luxi Yang. ConvTransformer: A convolutional transformer network for video frame synthesis. arXiv preprint arXiv:2011.10185, 2020.

---

> ### Author Response · Authors · 2024-12-01
> **To reviewer iPfR:  Enrich the Related Work (Part1). (Further Revision)**
>
> Dear reviewer iPfR,
>
> We sincerely thank you for acknowledging our first revision addressed the concerns regarding W1 and W3. We also would like to express our heartfelt appreciation to the reviewer for raising the score. Indeed, as you mentioned, the discussions about Transformer-based methods and GAN-based methods are still insufficient due to the page limitation. We appreciate your valuable suggestions for adding these insightful methods, including ConvTransformer [Ref.10], TCTN [Ref.11], CLGAN [Refs.14,15], UA-GAN [Ref.20],  and the method proposed by Luo et al. [Ref.16],  into our related work. These methods could definitely enrich the related work section and the background of our manuscript. Therefore, in response to your main concern of W2, we have further expanded our related work on **PRECIPITATION NOWCASTING**, which could be directly found as follows. Moreover, the revised related work will also be added to the future version of our paper:
>
> Notable progress has been achieved by applying deep learning in precipitation nowcasting [Refs.1-5]. The initial attempts are deterministic methods focusing on spatiotemporal modeling. Researchers explore different spatiotemporal modeling structures such as RNN [Refs.6,7], CNN [Refs.8,9], and Transformer [Refs.10,11,12]. **Specifically, ConvTransformer [Refs.10] integrates convolution layers into the Transformer for better spatiotemporal modeling. TCTN [Refs.11] utilizes 3D-temporal convolutions to improve short-term dependencies in Transformer-based spatiotemporal learning. EarthFormer [Refs.12] is proposed to capture spatial-temporal features in earth system evolution by cuboid attention.** However, these methods have a shortage of blurry predictions, which hamper the nowcasting of extreme events. Probabilistic methods are proposed to alleviate blurriness [Refs.13,14,15,16,17,18]. **DGMR [Refs.13] applies GAN to produce realistic and spatio-temporally consistent predictions to reduce blurriness. CLGAN [Refs.14, 15] introduces an adversarial network with a novel long short-term memory to improve the nowcasting skills of heavy precipitation events. Luo et al. [Ref.16] conducted a comprehensive experimental study on kinds of GAN models in precipitation nowcasting. These GAN-based methods significantly increase the similarity between the predictions and ground truth, but the unstable training, and inaccuracy of position and shape [Refs.19] hinder the further improvement on short-term forecasting.** To further enhance precipitation nowcasting with accurate global movements, later methods combine blurry predictions with probabilistic models. **Two-stage UA-GAN[Refs.20]**,  DiffCast [Refs.19], and CasCast [Refs.21] exploit how to generate small-scale weather pattern conditioning on the blurry predictions. Although these deterministic-probabilistic coupling methods achieve both global accuracy and local details, they suffer from repeating training for different contexts and require blurry predictions as data for training. From the perspective of deblur, Our method is proposed to simplify the complex training process and enhance the generality for wide use in different contexts.

---

> > ### Author Response · Authors · 2024-12-01
> > **To reviewer iPfR: Enrich the Related Work (Part2). (Further Revision)**
> >
> > [Ref.1] Kang Chen, et al. Fengwu: Pushing the skillful global medium-range weather forecast beyond 10 days lead. arXiv preprint arXiv:2304.02948, 2023.
> >
> > [Ref.2] Tao Han, et al. Fengwu-ghr: Learning the kilometer-scale medium-range global weather forecasting. arXiv preprint arXiv:2402.00059, 2024b.
> >
> > [Ref.3] Wanghan Xu, et al. Extremecast: Boosting extreme value prediction for global weather forecast. arXiv preprint arXiv:2402.01295, 2024.
> >
> > [Ref.4] Tao Han, et al. Weather-5k: A large-scale global station weather dataset towards comprehensive time-series forecasting benchmark. arXiv preprint arXiv:2406.14399, 2024a.
> >
> > [Ref.5] Junchao Gong, et al. Weatherformer: Empowering global numerical weather forecasting with space-time transformer, 2024. arXiv preprint arXiv:2409.16321, 2024.
> >
> > [Ref.6] Xingjian Shi, et al. Convolutional lstm network: A machine learning approach for precipitation nowcasting. Advances in Neural Information Processing Systems, 28, 2015.
> >
> > [Ref.7] Yunbo Wang, et al. Predrnn: A recurrent neural network for spatiotemporal predictive learning. IEEE Transactions on Pattern Analysis and Machine Intelligence, 45(2):2208–2225, 2022.
> >
> > [Ref.8] Zhangyang Gao, et al. Simvp: Simpler yet better video prediction. In Proceedings of the IEEE/CVF Conference on Computer Vision and Pattern Recognition, pp. 3170–3180, 2022a.
> >
> > [Ref.9] Cheng Tan, et al. Temporal attention unit: Towards efficient spatiotemporal predictive learning. In Proceedings of the IEEE/CVF Conference on Computer Vision and Pattern Recognition, pp. 18770–18782, 2023.
> >
> > **[Ref.10] Zhouyong Liu, et al. ConvTransformer: A convolutional transformer network for video frame synthesis. arXiv preprint arXiv:2011.10185, 2020.**
> >
> > **[Ref.11] Ziao Yang, et al. TCTN: A 3D-temporal convolutional transformer network for spatiotemporal predictive learning. arXiv preprint arXiv:2112.01085, 2021.**
> >
> > **[Ref.12] Zhihan Gao, et al. Earthformer: Exploring space-time transformers for earth system forecasting. Advances in Neural Information Processing Systems, 35:25390–25403, 2022b.**
> >
> > **[Ref.13] Suman Ravuri, et al. Skilful precipitation nowcasting using deep generative models of radar. Nature, 597(7878):672–677, 2021.**
> >
> > **[Ref.14] Yan Ji, et al. CLGAN: a generative adversarial network (GAN)-based video prediction model for precipitation nowcasting. Geoscientific Model Development, 16(10):2737–2752, 2023.**
> >
> > **[Ref.15] Yan Ji, et al. CLGAN: A GAN-based video prediction model for precipitation nowcasting. EGUsphere, 2022:1–23, 2022.**
> >
> > **[Ref.16] Chuyao Luo, et al. Experimental study on generative adversarial network for precipitation nowcasting. IEEE Transactions on Geoscience and Remote Sensing, 60:1–20, 2022.**
> >
> > [Ref.17] Zhihan Gao, et al. Prediff: Precipitation nowcasting with latent diffusion models. Advances in Neural Information Processing Systems, 36, 2024.
> >
> > [Ref.18] Zewei Zhao, et al. Advancing realistic precipitation nowcasting with a spatiotemporal transformer-based denoising diffusion model. IEEE Transactions on Geoscience and Remote Sensing, 2024.
> >
> > [Ref.19] Demin Yu, et al. Diffcast: A unified framework via residual diffusion for precipitation nowcasting. In Proceedings of the IEEE/CVF Conference on Computer Vision and Pattern Recognition, pp. 27758–27767, 2024.
> >
> > **[Ref.20] Liujia Xu, et al. Two-stage UA-GAN for precipitation nowcasting. Remote Sensing, 14(23):5948, 2022.**
> >
> > [Ref.21] Junchao Gong, et al. Cascast: Skillful high-resolution precipitation nowcasting via cascaded modelling. In Forty-first International Conference on Machine Learning.
> >
> > Thank you once again for your recognition of our work as well as our first revision and the valuable time you have invested in this review. Your suggestions could definitely improve the overall quality of our paper. We look forward to receiving your important feedback.
> >
> > Best regards,
> >
> > The Authors

---

> ### Comment · Reviewer_iPfR · 2024-12-01
>
> Thank you for the thoughtful revision and for addressing my comments regarding W2.   I have raised my score accordingly.
>
> Based on my understanding, these GAN-based methods, such as DGMR, CLGAN, and Two-stage UA-GAN, are also addressing the challenge of improving nowcasting clarity. In the **related work section**, it would be helpful to discuss how your method compares to GAN-based approaches under the context of nowcasting clarity, specifically highlighting the advantages of your approach.
>
> Additionally, I suggest that the **Introduction** briefly mention this comparison as well, to provide a clearer context for your contributions right from the beginning. I hope the authors can consider these points in future revisions to further strengthen the manuscript.

---

> > ### Author Response · Authors · 2024-12-02
> > **Thank You for Raising Score**
> >
> > Dear reviewer iPfR,
> >
> > Thank you for immediately reading our further revision on related work and raising your score. We will follow your insightful suggestion of specifically highlighting the advantage of superior generalization of our method compared with others, such as DGMR, CLGAN and Two-stage UA-GAN in both related work and introduction to further strengthen the manuscript. Thank you once again for your positive rating.
> >
> > Best regards,
> > The Authors

---

> > > ### Author Response · Authors · 2024-12-03
> > > **Official Comment by Authors**
> > >
> > > Thank you for your recognition of our work. We have provided General Comments and a Revision Summary, and we appreciate your constructive feedback that has helped make our paper better. We also welcome your participation in further discussions regarding the General Comments.

---

### Official Review · Reviewer_SepJ · 2024-11-03

**Soundness:** 2
**Presentation:** 2
**Contribution:** 2
**Rating:** 3
**Confidence:** 3

**Summary:**

The paper proposed a universal post-processing method, PostCast, designed for precipitation nowcasting. This method achieves denoising of convolution based prediction models using unconditinal DDPM through two key innovative points: zero shot blur estimation mechanism and auto scale gradient guidance strategy. The model was trained on a combination of five different datasets and prediction results generated by different models, and outperforms other conditional denoising models in terms of CSI metrics.

**Strengths:**

1. The paper proposes a novel approach to estimating blurriness by utilizing the gradient of metric respect to kernel parameters, which serves as a guide for the sampling of x_ {t-1}.
2. The paper introduces an auto-scale gradient guidance strategy that automatically calculates the guidance scale corresponding to different forecast time periods, models, and datasets. This strategy enables the model to adaptively denoise the prediction results.

**Weaknesses:**

1. CasCast and DiffCast are both designed for **predicting precipitation**, but not **postprocessing tasks**. It is crucial for the authors to compare with other generative models such as GAN. This would provide a more comprehensive evaluation of PostCast's performance against a wider range of methodologies.
2. The authors have compared PostCast with CasCast and DiffCast only in **out-of-distribution datasets** but did not compare them on HKO7, SEVIR, etc.
3. Relying on CSI as the only metric may not sufficiently judge the quality of precipitation prediction. The average intensity of the precipitation predictions can significantly impact CSI. For instance, using simple histogram matching on predictions can also significantly improve CSI. The authors should clarify whether the CSI improvement is due to **increased intensity** from the model or other factors to provide a more accurate assessment of the model's performance.

**Questions:**

The questions here are related to the three points described as weaknesses.

---

> ### Author Response · Authors · 2024-11-29
> **Answer of W1 To  Reviewer SepJ**
>
> We appreciate the reviewer’s constructive and detailed feedback. We try to address the concerns and questions below.
> > **W1:  Comparisons with other generative models for postprocessing tasks.**
>
>
> **A1**: We thank you for the insightful comment. We agree that comparing our PostCast with other generative models such as GAN is crucial. Therefore, following your advice, we have compared our PostCast with **a GAN-based postprocessing method** (DGP[Ref.1]) and **a diffusion-based postprocessing method** (GDP[Ref.2]) in our revised manuscript. The results are presented in Table 1 and Table 3 in our revised paper. We also show a part of the results in the tables below for convenience:
>
> ### SEVIR
> | model   | P1   | P4   | P16   | CSI-avg.   | HSS-avg.   | POD-avg.   |
> |-------|-------|-------|-------|-------|-------|-------|
> | EarthFormer | 0.032 | 0.024 |0.023 | 0.288 |0.374 | 0.357 |
> | +GDP | 0.001 | 0.002 |0.007 | 0.128 |0.159 | 0.191 |
> | +DGP | 0.020 | 0.042 |0.070 | 0.284 |0.372 | 0.355 |
> | +ours | 0.045 | 0.070  |0.131 | 0.308 |0.403| 0.427 |
>
>
> ### HKO7
> | model   | P1   | P4   | P16   | CSI-avg.   | HSS-avg.   | POD-avg.   |
> |-------|-------|-------|-------|-------|-------|-------|
> | EarthFormer | 0.025 | 0.025 |0.035 | 0.270 |0.390| 0.334 |
> | +GDP |0.003|0.011|0.038|0.046|0.078|0.048|
> | +DGP |0.039|0.083|0.187|0.254|0.372|0.328|
> | +ours |0.066|0.125|0.257|0.270|0.392|0.395|
>
> ### TAASRAD19
> | model   | P1   | P4   | P16   | CSI-avg.   | HSS-avg.   | POD-avg.   |
> |-------|-------|-------|-------|-------|-------|-------|
> | EarthFormer |0.019|0.021|0.028|0.169|0.266|0.204|
> | +GDP | 0.002|0.004|0.021|0.029|0.053|0.031|
> | +DGP | 0.018|0.041|0.094|0.149|0.238|0.196|
> | +ours |0.044|0.067|0.143|0.182|0.286|0.283|
>
> ### Shanghai
> | model   | P1   | P4   | P16   | CSI-avg.   | HSS-avg.   | POD-avg.   |
> |-------|-------|-------|-------|-------|-------|-------|
> | EarthFormer |0.021|0.029|0.055|0.205|0.304|0.253|
> | +GDP |0.007|0.012|0.031|0.046|0.078|0.048|
> | +DGP |0.029|0.070|0.160|0.186|0.282|0.271|
> | +ours |0.048|0.098|0.226|0.217|0.321|0.396|
>
> ### SRAD2018
> | model   | P1   | P4   | P16   | CSI-avg.   | HSS-avg.   | POD-avg.   |
> |-------|-------|-------|-------|-------|-------|-------|
> | EarthFormer |0.036|0.034|0.040|0.201|0.311|0.244|
> | +GDP |0.004|0.009|0.027|0.067|0.111|0.074|
> | +DGP |0.044|0.089|0.176| 0.189|0.298|0.239|
> | +ours |0.095|0.155|0.276|0.217|0.321|0.396|
>
> ### SCWDS CAP30
> | model   | P1   | P4   | P16   | CSI-avg.   | HSS-avg.   | POD-avg.   |
> |-------|-------|-------|-------|-------|-------|-------|
> | EarthFormer | 0.021|0.024|0.036|0.210|0.298|0.258|
> | +GDP | 0.001  | 0.002 |0.009 | 0.023 |0.036 | 0.023|
> | +DGP | 0.027 | 0.059 |0.111 | 0.207 |0.294 | 0.258 |
> | +ours | 0.070 | 0.117  | 0.241 | 0.249 |0.350| 0.404 |
>
> ### SCWDS CR
> | model   | P1   | P4   | P16   | CSI-avg.   | HSS-avg.   | POD-avg.   |
> |-------|-------|-------|-------|-------|-------|-------|
> | EarthFormer | 0.072|0.065|0.063|0.315|0.417|0.406|
> | +GDP | 0.001 | 0.002 |0.003 | 0.308 |0.069 | 0.049 |
> | +DGP | 0.071 | 0.083 |0.099| 0.047 |0.409 | 0.397 |
> | +ours | 0.141 | 0.211  | 0.326 | 0.338 |0.444| 0.570 |
>
> ### MeteoNet
> | model   | P1   | P4   | P16   | CSI-avg.   | HSS-avg.   | POD-avg.   |
> |-------|-------|-------|-------|-------|-------|-------|
> | EarthFormer | 0.000  | 0.003 |0.008 | 0.175 |0.259| 0.219 |
> | +GDP | 0.006 | 0.014 |0.048| 0.047 |0.077 | 0.049 |
> | +DGP | 0.037 | 0.082 |0.186 | 0.167 |0.250 | 0.218 |
> | +ours | 0.019 | 0.058  | 0.164  |0.194 |0.287| 0.320 |
>
> As shown in these comprehensive comparisons, our PostCast has superior performance. Specifically, the predictions are generated by EarthFormer for these three postprocessing methods. Our PostCast significantly surpasses the GDP and GDP in the meteorological metrics CSI-P1, CSI-P4, CSI-P16, CSI-avg, HSS-avg, and POD-avg, which explains the effectiveness of specifical designs in our PostCast for the postprocessing in precipitation nowcasting.
>
> [Ref.1] Xingang Pan, Xiaohang Zhan, Bo Dai, Dahua Lin, Chen Change Loy, and Ping Luo. Exploiting deep generative prior for versatile image restoration and manipulation. IEEE Transactions on Pattern Analysis and Machine Intelligence, 44(11):7474–7489, 2021.
>
> [Ref.2] Ben Fei, Zhaoyang Lyu, Liang Pan, Junzhe Zhang, Weidong Yang, Tianyue Luo, Bo Zhang, and Bo Dai. Generative diffusion prior for unified image restoration and enhancement. In Proceedings of the IEEE/CVF Conference on Computer Vision and Pattern Recognition, pp. 9935–9946, 2023.

---

> ### Author Response · Authors · 2024-11-29
> **Answer of W2 To Reviewer SepJ**
>
> > **W2:  Comparisons with CasCast and DiffCast on in-domain datasets.**
>
>
> **A2**: Thanks for your valuable suggestion. To fulfill the comprehensive evaluation of our PostCast, we have added comparisons with CasCast and DiffCast in Table 1 (**in-domain datasets**) in our revised paper. A part of the results is also presented below:
>
> ### SEVIR
> | model          | P1    | P4    | P16   | CSI↑ | HSS↑  | POD↑  |
> |----------------|-------|-------|-------|------|-------|-------|
> | EarthFormer    | 0.032 | 0.024 | 0.023 | 0.288   | 0.374 | 0.357 |
> | CasCast        | 0.039 | 0.067 | 0.156 | 0.246   | 0.335 | 0.422 |
> | DiffCast       | 0.049 | 0.070 | 0.186 | 0.270   | 0.362 | 0.378 |
> | PostCast(ours) | 0.045 | 0.070 | 0.131 | 0.308   | 0.403 | 0.427 |
>
>
> ### HKO7
> | model          | P1    | P4    | P16   | CSI↑ | HSS↑  | POD↑  |
> |----------------|-------|-------|-------|------|-------|-------|
> | EarthFormer    | 0.025 | 0.025 | 0.035 | 0.270   | 0.390 | 0.334 |
> | CasCast        | 0.054 | 0.108 | 0.235 | 0.232   | 0.343 | 0.454 |
> | DiffCast       | 0.061 | 0.113 | 0.255 | 0.263   | 0.385 | 0.375 |
> | PostCast(ours) | 0.066 | 0.125 | 0.257 | 0.270   | 0.392 | 0.395 |
>
> ### TAASRAD19
>
> | model          | P1    | P4    | P16   | CSI↑ | HSS↑  | POD↑  |
> |----------------|-------|-------|-------|------|-------|-------|
> | EarthFormer    | 0.019 | 0.021 | 0.028 | 0.169   | 0.266 | 0.204 |
> | CasCast        | 0.040 | 0.064 | 0.128 | 0.136   | 0.221 | 0.301 |
> | DiffCast       | 0.044 | 0.076 | 0.174 | 0.167   | 0.267 | 0.260 |
> | PostCast(ours) | 0.044 | 0.067 | 0.143 | 0.182   | 0.286 | 0.283 |
>
> ### Shanghai
> | model          | P1    | P4    | P16   | CSI↑ | HSS↑  | POD↑  |
> |----------------|-------|-------|-------|------|-------|-------|
> | EarthFormer    | 0.021 | 0.029 | 0.055 | 0.205   | 0.304 | 0.253 |
> | CasCast        | 0.034 | 0.068 | 0.167 | 0.171   | 0.259 | 0.336 |
> | DiffCast       | 0.050 | 0.097 | 0.218 | 0.205  | 0.309 | 0.282 |
> | PostCast(ours) | 0.048 | 0.098 | 0.226 | 0.217  | 0.321 | 0.396 |
>
> ### SRAD2018
> | model          | P1    | P4    | P16   | CSI↑ | HSS↑  | POD↑  |
> |----------------|-------|-------|-------|------|-------|-------|
> | EarthFormer    | 0.036 | 0.034 | 0.040 | 0.201   | 0.311 | 0.244|
> | CasCast        | 0.061 | 0.109 | 0.240 | 0.167   | 0.269 | 0.333|
> | DiffCast       | 0.071 | 0.124 | 0.257 | 0.198   | 0.313 | 0.307|
> | PostCast(ours) | 0.095 | 0.155 | 0.276 | 0.217   | 0.321 | 0.396|
>
> The predictions to be post-processed are generated by EarthFormer. When the test datasets are included in the training domain, our PostCast still has significant advantages over the CasCast and DiffCast, demonstrating the strong generalization capability of PostCast across different datasets.

---

> ### Author Response · Authors · 2024-11-29
> **Answer of W3 To Reviewer SepJ**
>
> > **W3:  The underlying reasons for CSI improvement.**
>
>
> ### SEVIR-POINT-DISTRIBUTION
> | model | R\[0,16) | R\[16,74) | R\[74,133) | R\[133,160) | R\[160,181) | R\[181,219)| R\[219, 255\] |
> |-------|-------|-------|-------|-------|-------|-------|-------|
> | GroundTruth   | 78.43% | 11.03% | 8.03% | 1.64% | 0.47% | 0.33% | 0.07%|
> | EathFormer | 74.71% | 16.17% | 7.62% | 1.22% | 0.20% | 0.08% | 0.00%
> | +ours     | 74.58% | 14.98% | 7.77% | 1.90% | 0.49% | 0.24% | 0.05% |
>
> ### SEVIR-CSI
> | model | CSI-16 | CSI-74 | CSI-133 | CSI-160 | CSI-181 | CSI-219|
> |-------|-------|-------|-------|-------|-------|-------|
> | EathFormer | 0.6673 | 0.5713 | 0.2567 | 0.1377 | 0.0836 | 0.0091
> | +ours     | 0.6581 | 0.5752 | 0.2759 | 0.1603 | 0.1233 | 0.0438
>
> Thank you for raising this insightful point.
> First, we chose the CSI score to evaluate our method because it is the most widely referenced by meteorological bureaus and precipitation nowcasting papers [Ref.1,Ref.2]. Additionally, to give a more sufficient evaluation, we have also evaluated all methods in terms of CSI-avg, HSS-avg, and POD-avg in Table 1, Table 3, and Table 5 in our revised paper. All these metrics are improved after applying our PostCast, indicating the increased quality of precipitation prediction.
> Then, to mine the underlying reasons for CSI improvement, we have analyzed the distribution of points. The table named SEVIR-POINT-DISTRIBUTION shows the proportion of points in SEVIR distributed across the ranges \[0, 16), \[16, 74), \[74, 133), \[133, 160), \[160, 181), \[181, 219), \[219, 255\]. Our post-processing method improves the alignment of EarthFormer's predictions with the actual distribution, particularly in high-threshold intervals such as \[160, 181), \[181, 219), and \[219, 255\]. Correspondingly, the CSI values in these intervals have shown significant improvements (CSI-160, CSI-181, CSI-219) as shown in the table named SEVIR-CSI. These results demonstrated that the CSI improvement comes from the better alignments with the distributions of ground truth and our predictions, rather than the **increased intensity**.
>
> [Ref.1] Zhang, Yuchen, et al. "Skilful nowcasting of extreme precipitation with NowcastNet." Nature 619.7970 (2023): 526-532.
>
> [Ref.2] Ravuri, Suman, et al. "Skilful precipitation nowcasting using deep generative models of radar." Nature 597.7878 (2021): 672-677.

---

> > ### Comment · Reviewer_SepJ · 2024-12-01
> >
> > Thank you to the authors for incorporating additional experiments. However, these experiments still do not fully address my concerns.
> >
> > - **Comparison with other generative models**: Besides Earthformer, the authors included a GAN-based postprocessing method and a diffusion-based postprocessing method. However, these two models are purely designed for computer vision, making the comparison unfair.
> >
> > - **In-domain dataset comparison**:  I appreciate the authors' effort to include comparisons on in-domain datasets, which provide additional results for evaluation.
> >
> > - **CSI experiments**: The precipitation statistics demonstrate that PostCast's output distribution aligns well with the ground truth. In the high-precipitation range, PostCast exhibits a larger proportion compared to Earthformer, suggesting that the CSI improvement is driven by an increase in precipitation intensity. The additional experiment does not sufficiently demonstrate that the improvement of CSI comes from improving more essential aspects, such as location accuracy.
> >
> > As a result, I will not change my score.

---

> ### Author Response · Authors · 2024-12-02
> **Reply remained concerns of Reviewer SepJ**
>
> Dear reviewer SepJ,
>
> Thank you for the valuable time and feedback you have invested in this review. We sincerely hope our further clarification could address your concerns:
> >W1:  Comparison with other generative models
>
>
> Since our method is the first postprocessing method for recovering local details and aligning distribution in this domain as the Reviewer iPfR mentioned in Strengths 2 "**the first of its kind in the domain of precipitation nowcasting**", we are only able to compare with most widely used GAN-based and Diffusion-based deblurring methods from Computer Vision. The results demonstrated the specific design of our method for tackling blurry predictions in precipitation nowcasting, since traditional CV methods cannot be directly applied in precipitation nowcasting.
> >W2: In-domain dataset comparison
>
>
> Thank you for recognizing our additional results for evaluation on in-domain datasets.
> >W3: Further experiments
>
>
> Thanks for your insightful suggestion. We would like to re-clarify the motivation that our method aims to provide an independent framework of the spatiotemporal prediction model and can be added on top of any prediction model with a unified model. Our PostCast aligns the blurry predictions better with the distribution of ground truth in a generalizable way. This improvement in precipitation nowcasting quality could also be reflected in the newly added HSS, POD, SSIM and PSNR. Besides increasing the intensities, our PostCast also refines the quality of details and shapes which could be referred to in Fig.14-17 in Appendix.10. Moreover, we also have evaluated FID[Ref.1] to further demonstrate quality improvement:
>
> | model       | FID↓  |
> | ------------|------------|
> | EarthFormer         | 50.1 |
> | +ours | 41.0|
>
> Additionally, as observed in Fig.11-13 in Appendix.10, both CasCast(ICML'24) and DiffCast(CVPR'24) have the same phenomenon of increased intensity, while our PostCast demonstrates stronger generalizations for improving the quality of precipitation nowcasting, which are evaluated by CSI and other newly added metrics. To prove this, we have conducted additional experiments on CasCast and DiffCast, shown as follows:
>
> ### SEVIR-POINT-DISTRIBUTION
> | model | R\[0,16) | R\[16,74) | R\[74,133) | R\[133,160) | R\[160,181) | R\[181,219)| R\[219, 255\] |
> |-------|-------|-------|-------|-------|-------|-------|-------|
> | GroundTruth   | 78.43% | 11.03% | 8.03% | 1.64% | 0.47% | 0.33% | 0.07% |
> | EathFormer | 74.71% | 16.17% | 7.62% | 1.22% | 0.20% | 0.08% | 0.00% |
> | +CasCast     | 81.31% | 8.82% | 6.69% | 1.67% | 0.61% | 0.49% | 0.40% |
> | +DiffCast     | 77.19% | 12.54% | 7.93% | 1.59% | 0.44% | 0.26% | 0.06% |
> | +ours     | 74.58% | 14.98% | 7.77% | 1.90% | 0.49% | 0.24% | 0.05% |
>
> ### COMPARISON on SEVIR
> | model          | P1    | P4    | P16   | CSI↑ | HSS↑  | POD↑  |
> |----------------|-------|-------|-------|------|-------|-------|
> | EarthFormer    | 0.032 | 0.024 | 0.023 | 0.288   | 0.374 | 0.357 |
> | CasCast        | 0.039 | 0.067 | 0.156 | 0.246   | 0.335 | 0.422 |
> | DiffCast       | 0.049 | 0.070 | 0.186 | 0.270   | 0.362 | 0.378 |
> | PostCast(ours) | 0.045 | 0.070 | 0.131 | 0.308   | 0.403 | 0.427 |
>
> Specifically, CasCast increases the most intensities in high thresholds among these three methods, and DiffCast best matches the distribution of intensity with ground truth. However, our PostCast significantly surpasses DiffCast and CasCast in CSI, HSS and POD, indicating the improvement of quality not only comes from increasing the intensity.  We will carefully discuss this common phenomenon in Discussion section.
>
> Thank you once again for your helpful constructions. We would greatly appreciate it if you could consider improving the evaluation after reviewing our responses. Thank you very much for your consideration!
>
> Best regards,
>
> Sincerely yours,
>
> Authors.
>
> [Ref.1 Heusel, Martin, et al. "Gans trained by a two time-scale update rule converge to a local nash equilibrium." Advances in neural information processing systems 30 (2017).]

---

> > ### Comment · Reviewer_SepJ · 2024-12-03
> >
> > I appreciate the authors’ explanation for why the comparison was limited to computer vision models and the additional experimental results on CSI. However, the explanation and results provided are not sufficiently convincing, and the two corresponding issues remain unresolved. Therefore, I will not raise my score.

---

> > > ### Author Response · Authors · 2024-12-03
> > > **Apply Reviewer SepJ**
> > >
> > > Thank you for your timely feedback.
> > > >W1:   First unsupervised generative postprocessing method
> > >
> > > We would like to emphasize that our PostCast is the first unsupervised generative postprocessing method for precipitation nowcasting, therefore we can not compare with other GAN-based or Diffusion-based postprocessing methods with the same type. This is also confirmed by Reviewer iPfR mentioned in Strengths 2 "**the first of its kind in the domain of precipitation nowcasting**".
> > >
> > > >W3:  More evaluations
> > >
> > >
> > > We have added metrics HSS, POD, SSIM, and PSNR besides CSI to validate the improvement of quality after applying our PostCast. We have further evaluated the Fraction Skill Score (FSS) in the table below to further demonstrate the improvement of the location accuracy [Refs.1,2]. As a result, our PostCast has the highest FSS scores across 4 spatial scales, proving the effectiveness of our PostCast.
> > >
> > > | model       | FSS-scale1↑  |FSS-scale4↑  |FSS-scale8↑  |FSS-scale16↑  |
> > > | ------------|------------|------------|------------|------------|
> > > | EarthFormer         | 0.2118 |0.2366|0.2571|0.2836|
> > > | +DiffCast | 0.2083 | 0.2442 | 0.2800 | 0.3330|
> > > | +PostCast |0.2032 |0.2387 |0.2720 |0.3197|
> > > | +ours|0.2302 |0.2651| 0.2952 |0.3339|
> > >
> > > We would greatly appreciate it if you could consider improving the evaluation after reviewing our responses. Thank you very much for your consideration!
> > >
> > > Best regards,
> > >
> > > Sincerely yours,
> > >
> > > Authors.
> > >
> > > [Ref.1] Lan, Z. H. A. N. G., et al. "FSS-based Evaluation on Monsoon Precipitation Forecasts in South China from Regional Models with Different Resolution." Journal of Tropical Meteorology 29.3 (2023): 301-311.
> > >
> > > [Ref.2] Yan, Chiu-Wai, et al. "Fourier Amplitude and Correlation Loss: Beyond Using L2 Loss for Skillful Precipitation Nowcasting." The Thirty-eighth Annual Conference on Neural Information Processing Systems.

---

### Author Response · Authors · 2024-11-29
**General repsponse to all reviewers**

We are very grateful to all the reviewers for their valuable comments and suggestions on this article.

---
We are glad to see the reviewers' recognition of our work.
- "The paper proposes a **novel** approach to estimate blurriness by utilizing the gradient of metric respect to kernel parameters. The paper introduces an auto-scale gradient guidance strategy that automatically calculates the guidance scale corresponding to different forecast periods, models, and datasets." (**Reviewer SepJ**)
- "**Originality**: The primary innovation of PostCast lies in employing an unconditional diffusion model (DDPM) to remove blurriness from precipitation predictions without relying on paired data...This unsupervised deblurring approach appears to be the first of its kind in precipitation nowcasting, demonstrating **creative thinking**. **Completeness of Experiments**: The paper provides a **well-designed experimental setup**, covering multiple datasets and forecast models to showcase the generalizability and robustness of PostCast." (**Reviewer iPfR**)
- "The problem is **well-motivated**. Given that this framework is independent of the spatiotemporal prediction model and can be added on top of any prediction model, it makes this work **very useful and widely applicable**. The results **strongly demonstrate the ability** of their proposed method. **The experiments are extensive** and prove PostCast’s effectiveness in recovering weather patterns across multiple prediction models, datasets, and lead times."  (**Reviewer CFdX**)

---
We have now addressed all weaknesses and questions raised by reviewers. Changes in the manuscript are highlighted in blue. Below, we summarise the major updates:
- For a comprehensive evaluation, we have included additional metrics such as CSI-avg, HSS-avg, POD-avg, SSIM, and PSNR. The details about how to calculate HSS and POD are added to the revised paper line 369-370. (**Reviewer SepJ, iPfR, CFdX**)
- To prove the effectiveness of our PostCast, we have added comparisons with other postprocessing methods including a diffusion-based and a GAN-based method. (**Reviewer SepJ**)
- We compare our method with other methods including postprocessing ones and precipitation nowcasting ones in both in-domain datasets and out-of-domain datasets. (**Reviewer SepJ**)
- To gain deep insight into the strength of our PostCast, we have provided a detailed analysis of the sources of CSI improvement. (**Reviewer SepJ**)
- To provide a more complete assessment of PostCast, we have conducted comparisons with two GAN-based precipitation nowcasting methods. (**Reviewer iPfR**)
- To better contextualize the background and contributions of PostCast, we have enriched the related work section to discuss GAN and Transformer methods in precipitation nowcasting. (**Reviewer iPfR**)
- To make a clearer clarification, we have reorganized the "method" section, creating subsections to explain the “zero-shot kernel estimation mechanism” and “auto-scale denoise guidance strategy” in more detail. (**Reviewer CFdX**)

---
Last but not least, thanks again to PCs, ACs, and all reviewers for their time and effort in reviewing.

---

### Author Response · Authors · 2024-12-03
**General Comments and Revision Summary**

We sincerely thank all the reviewers for their insightful reviews and helpful suggestions, as well as for acknowledging the importance, comprehensiveness, and contributions of our work.

---
We would like to re-emphasize the main contribution that our method aims to provide an independent framework that can be applied to a large range of spatiotemporal models, prediction steps, and datasets. Our PostCast improves the modeling of small-scale weather patterns in the blurry predictions and aligns them better with the distribution of ground truth in a generalizable way. Our contribution is also recognized by Reviewer CFdX that "it makes this work **very useful and widely applicable**; The results **strongly** demonstrate the ability of their proposed method. ", and by Reviewer iPfR "**Originality...the first of its kind in the domain of precipitation nowcasting**; a **well-designed** experimental setup, covering multiple datasets and forecast models to showcase the **generalizability and robustness**".

---
We have submitted a revised version of our paper and provide a summary of the revisions below:

---
1. Based on the suggestions from **Reviewers SepJ, iPfR, CFdX**, we have included additional metrics such as CSI-avg, HSS-avg, POD-avg, SSIM, and PSNR for a more comprehensive evaluation and demonstrated better results of predictions.
2. Following **Reviewer SepJ's** feedback, we have also included comparisons with other postprocessing methods including a diffusion-based and a GAN-based method. As **Reviewer iPfR** confirmed that in Strengths 2 "**the first of its kind in the domain of precipitation nowcasting**". We are only able to transfer diffusion-based and GAN-based postprocessing methods from computer vision for comparison.
3. As suggested by **Reviewer iPfR** and encouraged by his expert in precipitation nowcasting, we have included GAN and Transformer methods in the revised related work and introduction.
4. **Reviewer iPfR** requested more comparisons with two GAN-based precipitation nowcasting methods. We conducted these experiments and included them as well as the corresponding discussions in the paper.
5. We have compared our method with other methods including postprocessing ones and precipitation nowcasting ones in both in-domain datasets and out-of-domain datasets as recommended by **Reviewer SepJ**.
6. Thanks to **Reviewer CFdX's** suggestions, we have re-organized the "method" section to make our paper easier to follow.
7. We have provided a detailed analysis of the sources of CSI improvement to **Reviewer SepJ**. Our CSI improvement not only comes from the location accuracy (FSS score) but also the better alignments with the distributions of ground truth and our predictions.

---
Lastly, we would like to express our gratitude once again to all the reviewers for their constructive feedback, which has greatly improved our paper. We also hope that the reviewers will be available for further review discussions.

---

### Meta-Review · Area_Chair_hXL7 · 2024-12-23

**Metareview:**

The authors present a method to postprocess blurry precipitation future predictions by using a fine-tuned unconditional diffusion model on precipitation data (to have a reasonable generative prior on precipitation images) and guiding its generation process to the accurate prediction by using a blurry prediction (from any spatiotemporal predictive model) and a blur kernel estimation optimization algorithm. The authors state their primary motivation for this method (as opposed to existing conditional GAN/diffusion models to produce sharp forecasts) as to avoid repeated training for new OOD data and simplifying the training workflow of such conditional models that require blurry predictions as part of the training data.

The authors show results on applying their method (called PostCast) on spatiotemporal predictions from different models that show that PostCast is very effective in generating sharp predictions, reflected in several metrics. They further compare with other generative based approaches and show that their method is superior.

Strengths: Effective postprocessing method with positive results (clear visuals and many metrics) for precipitation nowcasting (an important research domain), experiments cover many datasets both in-domain and out-of-domain.
Weaknesses: The main weakness that seems to have been raised is the incomplete contextualization of the significance of the results/method. As a result, the contribution may be viewed as an interesting alternative to precipitation nowcasting through generative models, but incremental.

Points of improvement relating to the weakness:
* One issue raised by a reviewer is in incomplete comparisons with GAN-based postprocessing models. It is unclear why the CV models are chosen as baselines for this purpose. More in-depth investigations into GAN-based postprocessing models can greatly strengthen the paper.
* Relating to the above major issue, another issue was raised: The reviewer is also concerned about the applicability of the postprocessing method. One motivation is to reduce training complexity of requiring blurry predictions as data (for conditioning the generative model). More detailed discussions could have clarified this motivation better. Example:
  - it is unclear if all benchmarked GAN models have this issue (DGMR for example - does it require blurry prediction as data?). If not, this
    needs to be discussed with proper context
  - PostCast is described as unsupervised - however, it seems that there is a finetuning stage with ground truth predictions from the
    training dataset. This makes it difficult to judge how the training complexity is reduced because (1) both PostCast and DiffCast (etc)
    require some blurry prediction model (same complexity) and (2) DiffCast requires blurry predictions as further input for the full dataset
    (inference runs) and PostCast requires finetuning an unconditional diffusion model . Currently, the paper reads as though the
    motivation is if the blurry prediction model is for some reason a black box. Further, models like DiffCast could also be trained with any
    backbone (spatiotemporal model).

What could help: A clearer discussion on (1) the value of PostCast (some quantitative characterization of the cost that makes "the training pipeline cumbersome" (from the abstract)) with (2) more details on how existing methods work exactly (DGMR vs DiffCast vs PostCast vs CV-based GAN models) and why they are good baselines; (3) a discussion on other GAN-based postprocessing methods (downscaling models, bias-correction models) and (4) presenting PostCast as a valuable alternative in light of (1) rather than as a new SOTA model.

**Additional Comments On Reviewer Discussion:**

Reviewers raised several issues:
* Incomplete literature review of GANs and other generative models and incomplete comparisons to such models - the authors addressed this by adding two CV models as well as two GAN based models. 2/3 reviewers agree that this is a good effort in adding appropriate comparisons. The authors also expanded their literature review with these models.
* Incomplete metrics - the authors added more metrics in addition to CSI
* Additional experiments for in-domain evaluation - the authors completed this and results still show some benefits of PostCast
* Clarity of methods section - the authors revised this section to be more clear in what the method is

After the rebuttal and internal discussions, the reviewers still did not reach a unanimous decision with one strong reject (SepJ), one strong accept (CFdX), and one neutral score (iPfR). SepJ is still concerned about the useful applicability of postprocessing the predictions as opposed to previous methods such as DiffCast as well as the soundness of the performance gains of PostCast. The other two reviewers feel the alternative methodology to achieve good deblurred predictions with sufficient experimental evidence is still a valuable contribution to the community, despite incremental improvements. I lean towards acceptance due to this.

---

### Decision · Program_Chairs · 2025-01-22

Accept (Poster)